# Geometric Collapse: When Vision Models Fail to Verify Physical Causality

Wentao Zhang [1]   Jinhu Qi [2 3]   Weiqiang Jin [4]   Yifei Zhang [5]   Chan-Tong Lam [1]   Irwin King [2]

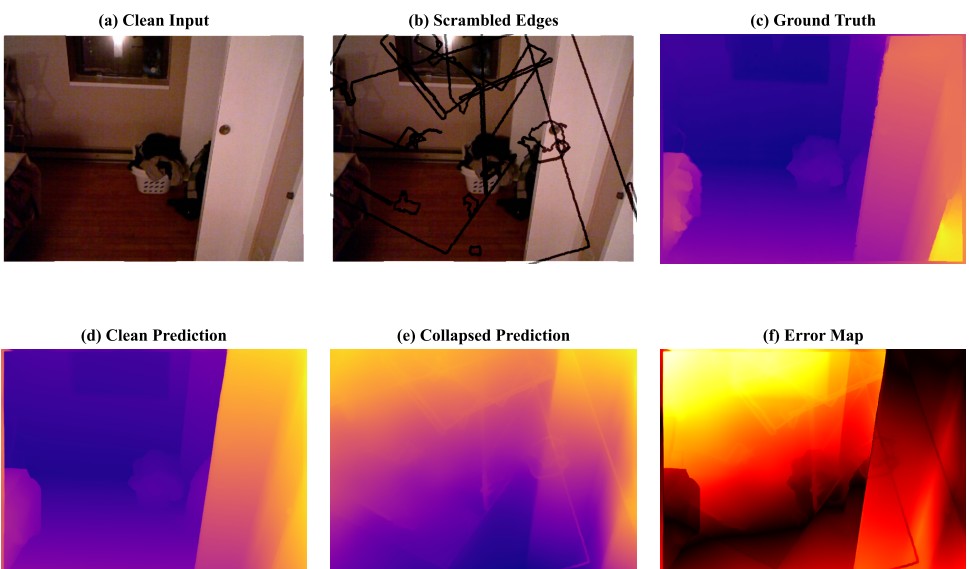

Figure 1. **Geometric Collapse: Dense models fail to verify physical consistency. Top:** "Scrambled Edges"—edge cues energy-matched to noise but violating continuity/illumination/occlusion priors—are treated as valid structure by dense predictors (e.g., DepthAnythingV2), causing global hallucinations (*Geometric Collapse*). **Bottom:** Models stable under frequency-matched noise can still collapse under unsupported edges, with errors propagating scene-wide.

## Abstract

Recent progress in large-scale self-supervised learning has improved dense geometric prediction, but it remains unclear whether such scaling yields inference-time physical plausibility checks. We propose *Scrambled Edges*, a controlled counterfactual that injects salient edge-like cues while violating surface continuity, illumination coherence, and occlusion ordering. With energy-matched and structure-matched controls, we isolate the effect of unsupported edge evidence from high-frequency energy and edge sparsity.

Across CNN/ViT/SSL depth predictors on NYU Depth v2 and KITTI, *Scrambled Edges* induce up to $3.2\times$ larger deviation from clean predictions than energy-matched noise; additional diffusion and flow-matching depth estimators show attenuated but still significant collapse. The resulting *Geometric Collapse* propagates globally: even with oracle knowledge of the corrupted region, output-level repair recovers only $47\%$, with substantial error outside the mask. These findings provide controlled behavioral evidence that current dense predictors lack reliable mechanisms to quarantine physically unsupported edge cues, motivating explicit plausibility scoring and selective cue integration.

[1]Faculty of Applied Sciences, Macao Polytechnic University, Macao SAR, China    [2]The Chinese University of Hong Kong, Hong Kong SAR, China    [3]AgentecFusion Limited    [4]School of Information and Communications Engineering, Xi'an Jiaotong University, Xi'an, Shaanxi, China    [5]School of Computer Science, Northwestern Polytechnic University, Xi'an, Shaanxi, China . Correspondence to: Chan-Tong Lam <ctlam@mpu.edu.mo>, Irwin King <king@cse.cuhk.edu.hk>.

*Proceedings of the $43^{rd}$ International Conference on Machine Learning*, Seoul, South Korea. PMLR 306, 2026. Copyright 2026 by the author(s).

## 1. Introduction

Modern dense vision models have improved rapidly with Vision Transformers (ViTs), large-scale self-supervised learning (SSL), and increasingly diverse training data. These advances have made monocular depth and surface normal

predictors much more accurate on standard benchmarks. Yet high average accuracy does not answer a more targeted question: *when a visually salient cue conflicts with basic physical structure, does the model reject the cue or fold it into its geometric prediction?*

We study this question through image *edges*. Edges are central to geometric inference because many strong intensity gradients correspond to physical events: object boundaries, occlusion contours, cast shadows, or material transitions. At the same time, not every edge is valid geometric evidence. Reflections, lighting changes, and other visual artifacts can produce strong local gradients without requiring a change in 3D layout. A robust dense predictor should therefore not merely detect edges; it should use them selectively, according to whether they are physically supported.

Our hypothesis is that modern dense predictors can treat edge-like cues as geometry-relevant signals without reliably verifying whether those cues admit a plausible physical explanation. If so, visually salient but physically unsupported edges should be adopted into the prediction and may trigger failures that extend beyond the perturbed pixels. We call this negative emergence at the behavioral level: scaling and SSL may improve benchmark accuracy without producing reliable inference-time physical verification.

To test this hypothesis, we operationalize physical support using three simple priors: *continuity*, where depth and normals should vary smoothly on a single surface; *illumination coherence*, where shading and shadow edges should be compatible with plausible lighting and geometry; and *occlusion causality*, where boundary evidence should admit a consistent depth ordering. These priors are not intended as a complete physics engine. They provide a controlled way to ask whether an edge-like cue has enough physical support to be trusted. Formal definitions and measurements appear in §3 and Appendix D.

We introduce **Scrambled Edges**, a controlled counterfactual intervention for this test. The protocol relocates and transforms real edge segments so that they remain visually salient but are unlikely to correspond to valid 3D structure. We compare these perturbations against energy-matched and structure-matched controls, separating physical-prior violations from generic high-frequency noise or edge sparsity. We then measure whether predictions remain stable, whether boundary structure degrades, and whether errors stay local or propagate globally. The goal is not merely to measure robustness, but to probe whether dense predictors suppress unsupported evidence before it shapes the output.

Throughout this paper, we focus on *observable behavior under controlled diagnostics*; when we refer to an "absence" of verification, we mean the absence of such behavioral evidence, not a claim about representational impossibility.

This setup leads to three research questions:

- **RQ1 (Adoption and prevalence).** Do dense predictors adopt visually salient but physically unsupported edges, and is this failure prevalent across CNN/ViT/SSL architectures—even when they are robust to energy-matched noise?

- **RQ2 (Mechanism and propagation).** Which physical-prior violations drive collapse, and does the induced error remain localized or propagate globally (imposing a hard ceiling on local repair)?

- **RQ3 (Evaluation validity).** Do standard global metrics expose this failure, and does it generalize across datasets, tasks, real long-tail ambiguities, and downstream geometric proxies?

Our main empirical finding is a global failure mode we term **Geometric Collapse**: local edge evidence that violates physical priors can cause dense predictors to lose coherent 3D structure, producing scene-wide deviations rather than localized artifacts. This behavior is distinct from ordinary noise sensitivity. Models can remain comparatively stable under energy-matched noise while changing substantially under unsupported edge cues. Even oracle knowledge of the perturbed region only partially repairs the output, showing that the error has already propagated beyond the manipulated pixels. Across models and datasets, these results suggest that scaling and pretraining alone do not guarantee reliable physical-causality verification in dense geometric reasoning.

**Contributions.**

- **Negative emergence and a falsification-style diagnostic.** We provide controlled behavioral evidence that scaling and SSL do not yield emergent *physical verification behavior*. We introduce *Scrambled Edges*, which violates continuity/illumination/occlusion priors while controlling frequency energy, with matched controls to isolate physical violation from high-frequency content.

- **Geometric Collapse: mechanism, propagation, and repair ceiling.** We identify a global failure mode across CNN/ViT/SSL predictors, revealing a paradox where noise robustness does not imply edge robustness. We disentangle prior contributions (occlusion causality dominates), show collapse propagates beyond perturbed pixels, and establish a hard oracle-repair ceiling due to spillover.

- **Evaluation insights and practical relevance.** We demonstrate a metric paradox (global GT metrics miss the failure; boundary-aware metrics reveal it), validate task selectivity and cross-dataset generalization, connect to real long-tail edge ambiguities, and quantify downstream geometric consequences.

## 2. Related Work

**Robustness diagnostics beyond statistical noise.** Robustness in vision is often studied through diagnostic corruptions and distribution shifts. ImageNet-C, for example, focuses on common statistical degradations (Hendrycks & Dietterich, 2019), while later analyses show sensitivity to frequency components (Yin et al., 2019; Wang et al., 2020) and small geometric shifts (Azulay & Weiss, 2019; Zhang, 2019). These results connect to shortcut learning, where models rely on predictive but non-causal cues such as texture rather than verified structure (Geirhos et al., 2019; 2020). In dense prediction, related robustness concerns include monocular depth under procedural or sensor-like perturbations (Nugent et al., 2025), boundary fidelity (Hu et al., 2018), and adaptation under distribution shift (Chen et al., 2021; Murez et al., 2017). Adversarial examples and patches reveal further fragility (Goodfellow et al., 2015; Brown et al., 2018), but their optimized patterns are often hard to interpret structurally. Our work instead intervenes on interpretable *edge evidence*, using energy-matched and structure-matched controls to isolate physical-support violations from generic high-frequency content.

**Physical priors and geometric consistency.** The view that perception should respect physical regularities has deep roots in classical vision and surface perception (Gibson, 1979; Marr, 1982; Nakayama & Shimojo, 1992; Kersten et al., 2004). Modern systems encode related constraints through geometric consistency objectives, such as photometric or left-right consistency for self-supervised monocular depth (Godard et al., 2017), and through inverse-graphics formulations that recover latent physical factors (Jaques et al., 2020). 3D-consistent representations such as NeRF and Gaussian Splatting similarly use multi-view coherence as a path toward physically grounded modeling (Mildenhall et al., 2020; Kerbl et al., 2023). These approaches impose physical structure through training objectives or explicit 3D representations. We ask a complementary behavioral question: during *single-image inference*, does an end-to-end dense predictor discount edge cues that are visually strong but physically unsupported?

**Modern monocular depth models, scaling, and evaluation.** Monocular depth estimation has advanced rapidly with stronger backbones and larger training corpora, including transformer-based dense prediction (Ranftl et al., 2020; 2021; Dosovitskiy et al., 2021) and scalable SSL pretraining (He et al., 2021; Oquab et al., 2024). Recent models improve transfer through zero-shot metric alignment or large-scale unlabeled training (Bhat et al., 2023; Yang et al., 2024), and foundation models can provide additional alignment signals (Kirillov et al., 2023). However, stronger average-case performance does not guarantee robustness under shift or adversarial conditions (Tripuraneni et al., 2021; Wicker & Kwiatkowska, 2019). Standard depth metrics also emphasize global pixel-wise error, while boundary fidelity and structural alignment require more targeted measurements (Hu et al., 2018). Complementary to local repair mechanisms such as inpainting-style methods (Telea, 2004), we show that unsupported edge evidence can trigger *global* geometric degradation and impose a hard ceiling on local repair. This motivates evaluation that probes inference-time physical verification directly, beyond scaling, pretraining, and global metrics.

## 3. Methodology

Our diagnostic framework is illustrated in Figure 2. We frame Scrambled Edges as a falsification-style, counterfactual test of *physical verification behavior*: we inject visually salient but geometrically unsupported cues (*Scrambled Edges*), compare model stability against energy-matched noise controls, and quantify the resulting Geometric Collapse across stability, structural, and downstream dimensions.

### 3.1. Scrambled Edges: Physically Unsupported Edge Cues

We introduce *Scrambled Edges*, a diagnostic perturbation that inserts visually edge-like cues while breaking physical 3D explanations. Given an input image $I \in \mathbb{R}^{H \times W \times 3}$, we first extract edge segments using Canny (Canny, 1986) and select the top-$K$ connected components by area (visualized in Figure 2, top-left).

**Definition 3.1** (Scrambled Edges). Let $E = \{e_1, \ldots, e_K\}$ denote the selected edge segments. For each segment $e_k$, we sample an affine transform $T_k = (\theta_k, t_k)$ with

$$\theta_k \sim \mathcal{U}(-60°, +60°), \tag{1}$$
$$t_k \sim \mathcal{U}(-0.25W, +0.25W) \times \mathcal{U}(-0.25H, +0.25H), \tag{2}$$

and warp the segment to obtain $T_k(e_k)$. Let $M_{\text{scram}} = \bigcup_{k=1}^{K} T_k(e_k)$ be the binary mask of scrambled edges. We form the perturbed image

$$I_{\text{scram}} = \text{clip}(I - \alpha \cdot M_{\text{scram}}, 0, 1), \tag{3}$$

where $\alpha \in [0, 1]$ controls perturbation intensity (assuming $I \in [0, 1]$).

**Defaults.** Unless stated otherwise, we use $K=15$ and $\alpha=0.8$, yielding $\sim 36\%$ mask coverage on NYU.

**Why this violates physical priors.** Translation places edge cues in geometrically smooth regions (violating surface continuity); darkening introduces contrast not explained by

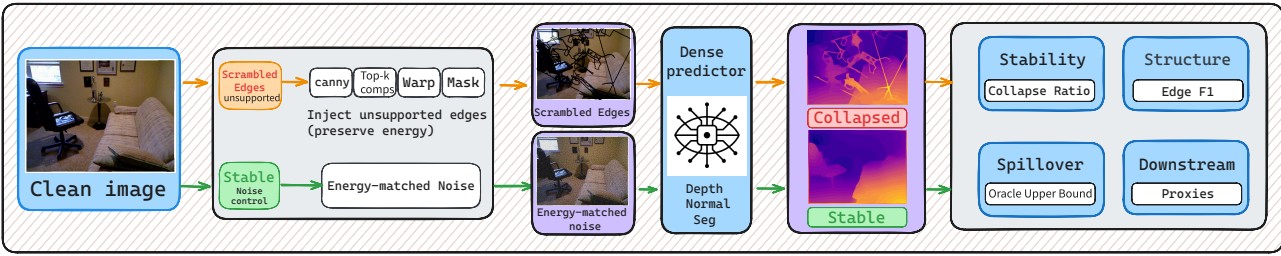

*Figure 2.* **Overview of the Scrambled Edges Diagnostic Pipeline.** We probe *physical-causality verification* by contrasting model behavior under two energy-matched conditions: (top) **Scrambled Edges**, which inject visually salient but geometrically unsupported edge cues, and (bottom) **High-pass Noise**, which serves as a frequency control. We quantify the resulting *Geometric Collapse* across four dimensions: **Stability** (Collapse Ratio), **Structure** (Edge F1), **Spillover** (Oracle repair limits), and **Downstream** geometric proxies.

*Table 1.* Scrambled Edges operations and the targeted support signals.

| Operation | Image-level change | Targeted support signal |
|---|---|---|
| Translation | Edges placed in geometrically smooth regions | Surface continuity (depth gradient alignment) |
| Darkening | Contrast without plausible lighting source | Illumination coherence (chromatic signature) |
| Rotation | Junction geometry incompatible after reorientation | Occlusion ordering (T-junction proxy compatibility) |

illumination (violating illumination coherence); rotation disrupts junction compatibility under our T-junction proxy for occlusion ordering. A concise mapping is given in Table 1, with full proxy validation in Appendix D.

### 3.2. Control Groups

We employ two rigorous control conditions to isolate the mechanism of collapse:

**1. Energy-Matched High-pass Noise (Frequency Control).** To separate *physical structure* from mere *high-frequency energy*, we compare against energy-matched high-pass noise (unsharp-mask style). We generate Gaussian white noise $\epsilon \sim \mathcal{N}(0, \sigma_n^2)$ and apply a high-pass filter to isolate edge-like frequencies:

$$I_{\text{noise}} = \text{clip}\big(I + (\epsilon - G_\sigma(\epsilon)), 0, 1\big), \quad (4)$$

where $G_\sigma$ is a Gaussian blur with $\sigma = 5$ pixels. The noise scale $\sigma_n$ is calibrated per-image such that the global Root Mean Square (RMS) amplitude of the perturbation matches that of the Scrambled Edges ($\text{RMS}(I_{\text{noise}} - I) \approx \text{RMS}(I_{\text{scram}} - I)$). Specifically, we compute $\sigma_n = \text{RMS}(I_{\text{scram}} - I)/\text{RMS}(\epsilon - G_\sigma(\epsilon))$ using a unit-variance noise sample, then scale accordingly. This control tests whether collapse is triggered by high-frequency content alone.

**2. Edge-Shaped Noise (Structure Control).** To test

whether the *shape* of edges alone causes collapse (without geometric violation), we generate *Edge-Shaped Noise*. We extract edge masks identical to Scrambled Edges but apply noise/color shifts *in place* without moving them (no rotation or translation).

$$I_{\text{edge}} = I + \text{Noise}(M_{\text{edge}}), \quad (5)$$

where $M_{\text{edge}}$ corresponds to the original edge locations (without relocation/rotation). This control preserves the "edge-like" visual statistics but avoids introducing geometric violations from relocation/rotation. Collapse under Scrambled Edges but not Edge-Shaped Noise confirms that *false position* (causal violation), not just edge presence, drives the failure.

### 3.3. Quantifying Physical Consistency of Edge Cues

We operationalize "physical support" by checking whether edge pixels align with ground-truth geometric discontinuities. Let $M_{\text{edge}}$ denote an edge mask (from Canny on the corresponding image). We define the **False Edge Ratio**:

$$R_{\text{false}} = \frac{\sum \big(M_{\text{edge}} \cdot \mathbb{I}(|\nabla D_{\text{gt}}| < \tau)\big)}{\sum M_{\text{edge}}}, \quad (6)$$

where $D_{\text{gt}}$ is the ground-truth depth map and $\tau$ is a threshold for "geometrically smooth" regions. We set $\tau$ to the median of $|\nabla D_{\text{gt}}|$ over NYU (reported in Appendix), and report **G-Score** $= 1 - R_{\text{false}}$ for readability.

**Geometry gap.** On NYU ($N$=1449), clean-image edges are substantially more likely to be geometrically supported (by depth discontinuities) than scrambled-edge masks. We report the exact G-Score statistics in Appendix D (Table 15). *Note:* many real image edges are photometric (texture/shadow/material) rather than depth discontinuities, so a low absolute G-Score for clean Canny edges is expected; we therefore treat geometry as a conservative proxy and report additional priors.

**Full prior validation.** Geometry alone is a conservative check. **Score definitions: G-Score** is the fraction of edge

pixels with depth gradient $> \tau$; **P-Score** is the fraction explained by shadow/texture chromatic signatures in CIE Lab; **O-Score** is the fraction of detected T-junctions with consistent depth ordering. We report unified **G/P/O** validation in Appendix D (Table 15).

### 3.4. Local Repair to Measure Global Spillover

To isolate whether errors are local or globally propagated, we define an **output-level oracle repair** using the known perturbation mask $M_{\text{scram}}$:

$$D_{\text{patch}} = D_{\text{scram}} \cdot (1 - M_{\text{scram}}) + D_{\text{clean}} \cdot M_{\text{scram}}, \quad (7)$$

where $D_{\text{clean}}$ and $D_{\text{scram}}$ are depth predictions from clean and scrambled inputs, respectively. If errors were purely local, $D_{\text{patch}}$ would match $D_{\text{clean}}$; residual error measures **global spillover**.

We also report oracle-mask repair and inpainting analyses to quantify the spillover ceiling (Appendix C.3).

**Scope.** Scrambled Edges is a *diagnostic* perturbation designed to probe physical consistency, not a realistic camera corruption. We discuss connections to real long-tail edge ambiguities in §5.

### 3.5. Experimental Setup and Metrics

**Dataset.** NYU Depth v2 labeled set (Silberman et al., 2012) ($N$=1449).

**Models.** MiDaS v2.1 (Ranftl et al., 2020), MiDaS DPT (Ranftl et al., 2021), DepthAnything v1/v2 (Yang et al., 2024) (DINOv2 (Oquab et al., 2024) pretraining).

**Stability metrics.** We measure deviation from the model's clean prediction (not ground-truth accuracy) to test whether the model *rejects* physically unsupported edge cues:

- **RMSE$_\Delta$**: RMSE between perturbed and clean predictions.

- **Collapse Ratio**: $\text{RMSE}_{\Delta,\text{scram}}/\text{RMSE}_{\Delta,\text{noise}}$.

- **Recovery**: relative reduction of RMSE$_\Delta$ after a defense.

We also evaluate structural collapse using surface normals derived directly from the predicted depth maps (via local gradients), validating whether depth derivative consistency degrades more than depth itself. We discuss why global GT metrics can be misleading for this failure mode in §4 and Appendix E.

## 4. Experiments

We evaluate Geometric Collapse on NYU Depth v2 (Silberman et al., 2012) ($N$=1449 RGB-D pairs) across represen-

tative CNN-, ViT-, and SSL-based dense predictors.

**Primary metric (stability).** Our primary measurements quantify deviation from the model's *clean prediction* (behavioral stability). We show that standard GT metrics can be misleading for this failure mode due to smoothing artifacts, whereas structural metrics (Edge F1) reveal the collapse (Table 4 and Appendix E). Detailed numerical results are provided in Appendix B (Table 8).

### 4.1. Main Results: Geometric Collapse

**(RQ1)** We first compare model stability under Scrambled Edges against matched controls and report collapse severity across CNN/ViT/SSL predictors.

**Key finding (negative emergence / SSL paradox).** As illustrated by the architectural comparisons in Table 2 (and detailed in Appendix Table 8), we observe a negative emergence result at the behavioral level: *even with large-scale SSL pretraining*, models can be highly stable under unstructured noise yet *more* sensitive to edge cues that lack geometric justification, leading to higher collapse ratios. We treat the cause as a hypothesis: objectives that encourage reconstructing all salient tokens may increase reliance on edge-like signals regardless of whether they are physically supported by underlying geometry, amplifying collapse under prior violations.

**Cross-paradigm check.** To test whether collapse is specific to one-shot feed-forward regressors, we additionally evaluate generative depth estimators in Appendix B.4. Marigold, a diffusion-based depth estimator, shows attenuated but statistically significant collapse ($1.55\times$, Cohen's $d$=0.88), and DepthFM, a flow-matching estimator, also shows significant collapse ($1.11\times$, $d$=0.25). These results suggest that iterative or generative inference can reduce sensitivity, but does not by itself constitute explicit physical-support verification.

**Adoption is a prerequisite for collapse (representative evidence).** To directly connect stability collapse to cue *adoption*, we measure an *Adoption Rate* that quantifies how often injected scrambled-edge pixels induce aligned depth-gradient discontinuities (definition and full sensitivity in Appendix B). For MiDaS DPT on NYU, weak perturbations ($\alpha$=0.2) show low adoption (6.2%) and correspondingly mild collapse ($1.37\times$), whereas once adoption rises (24.7% at $\alpha$=0.6) the collapse crosses $2.0\times$ ($2.02\times$), and at the default setting ($\alpha$=0.8) adoption reaches 36.1% with collapse $2.34\times$ (Appendix Table 10).

*Table 2.* **Collapse ratios on NYU ($N{=}1449$), normalized by high-pass noise.** *Mask-Matched* applies unstructured noise within $M_{\text{scram}}$. Direction (occlusion causality) drives the strongest collapse.

| Condition (Prior Violation) | Collapse Ratio (RMSE$_\Delta$ / RMSE$_{\Delta,\text{high-pass}}$) | | | |
| --- | --- | --- | --- | --- |
| | MiDaS v2.1 (CNN) | MiDaS DPT (ViT) | DepthAnything v1 (SSL) | **DepthAnything V2** |
| **1. Baseline** | | | | |
| High-pass Noise (No Violation) | 1.00× | 1.00× | 1.00× | 1.00× |
| Edge-Shaped (Structure Only) | 1.00× | 1.31× | 1.01× | 1.71× |
| Mask-Matched (Position Only) | 0.58× | 0.60× | 0.43× | 0.53× |
| **2. Single Violation** | | | | |
| Darkening (Illumination) | 0.85× | 1.06× | 1.09× | 1.95× |
| T-Junctions (Local Occlusion) | 0.73× | 1.07× | 1.08× | 2.00× |
| Position (Continuity) | 1.22× | 1.68× | 1.68× | 3.15× |
| Direction (Causality) | 1.86× | **2.22×** | **1.99×** | **3.18×** |
| **3. Full Collapse** | | | | |
| **Scrambled Edges (All)** | 1.88× | 2.34× | 2.02× | 3.20× |

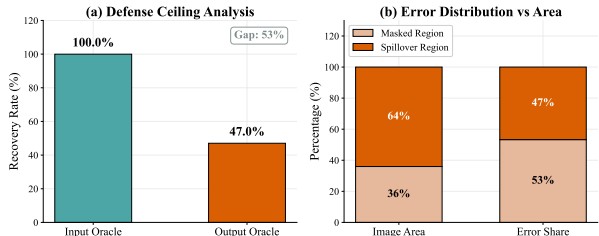

*Figure 3.* **The spillover limit.** Recovery $= 1 -$ RMSE$_{\text{defended}}$/RMSE$_{\text{undefended}}$. Local output repair is capped because error propagates beyond the perturbed region. See Appendix Table 13.

## 4.2. Mechanism: Prior-Violation Ladder

**(RQ2)** We ablate individual physical-prior violations to identify which cues dominate collapse along a controlled ladder.

**Key finding (structure vs. causality).** Table 2 presents a comprehensive mechanism analysis across all four architectures (visual comparison in Appendix Fig. 9). The results consistently validate the "Prior-Violation Ladder":

1. **Structure is benign:** Across all models, Edge-Shaped noise (structure without violation) induces minimal collapse (ratios 1.0×–1.7×) compared to the baseline.

2. **Causality is dominant:** The Direction perturbation disrupts occlusion causality and drives the strongest collapse across all models:
   2.22× (MiDaS ViT); 1.99× (DepthAnything v1); 3.18× (DepthAnything V2).

3. **V2 Sensitivity:** DepthAnything V2 is exceptionally sensitive, showing high collapse ($> 3.0\times$) whenever *any* prior is violated, suggesting its high performance comes at the cost of over-reliance on priors.

## 4.3. Defense Analysis: The Spillover Limit

**(RQ2)** We test whether collapse is local via oracle patch repair and quantify the spillover and repair ceiling.

**Key finding (global propagation).** The results in Fig. 3 reveal that collapse is not confined to the corrupted pixels: it propagates globally, creating a hard ceiling for local repair strategies. This is evidenced by the persistent gap between input-oracle and output-oracle recovery, where even perfect mask knowledge fails to eliminate hallucinations in unperturbed regions. We also test a simple post-hoc edge-consistency defense in Appendix C.3; it improves recovery but remains far below full repair, supporting the need for cue-selection mechanisms before geometric integration.

## 4.4. Multi-View Geometric Consistency

**(RQ2)** We provide a physically-grounded validation of spillover by measuring cross-view reprojection consistency.

To test whether the observed collapse translates to violations of 3D geometric constraints, we evaluate multi-view photometric consistency on KITTI Odometry (Geiger et al., 2012) (Seq 00/02/05; 750 frame pairs). For each condition $c \in \{\text{clean}, \text{scram}, \text{noise}\}$, we predict $D_t^c$ from the corresponding input $I_t^c$ while keeping intrinsics and the relative pose $T_{t \to t+1}$ fixed (using provided odometry poses, direction sanity-checked). To avoid RGB-corruption confounds, photometric error is computed using clean RGB only: we project pixels from frame $t$ into frame $t+1$ using $D_t^c$ and bilinearly sample $I_{t+1}^{\text{clean}}$ at the projected locations to reconstruct $I_t^{\text{clean}}$. We report mean $\ell_1$ error over valid projections, restricted to pixels outside $M_{\text{scram}}$ (the injected-edge mask on frame $t$).

**Key finding (multi-view metric paradox).** As shown in Table 3, photometric reprojection error is *not* a reli-

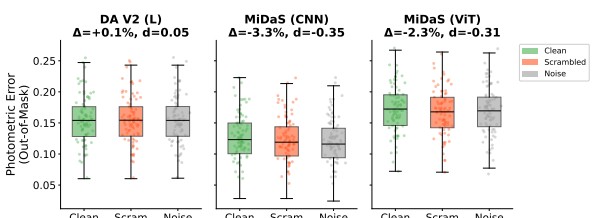

*Figure 4.* **Multi-view photometric consistency (out-of-mask).** MiDaS shows *decreased* photometric error under Scrambled Edges, which might suggest improved *appearance-based* consistency—but see Table 3 for the paradox: depth–depth consistency *degrades* sharply.

*Table 3.* **Multi-view metric paradox on KITTI Odometry.** Photometric error can *decrease* while depth–depth consistency error *increases* under Scrambled Edges, showing that appearance-based metrics can mislead when geometric structure collapses. Scale alignment applied consistently across conditions; depth consistency is mean relative error ($\times 100$; warped $D_t$ vs. sampled $D_{t+1}$) and is heavy-tailed—we focus on paired deltas.

| | Photometric ($\ell_1$) | | | Depth Consist. (rel.) | | |
|---|---|---|---|---|---|---|
| **Model** | Clean | $\Delta_s(\%)$ | $\Delta_n(\%)$ | Clean | $\Delta_s(\%)$ | $\Delta_n(\%)$ |
| DA V2 (L) | 0.1537 | +0.1% | +0.2% | 148.4 | −12.9% | −20.6% |
| MiDaS (ViT) | 0.1705 | −2.3% | −1.7% | 160.2 | **+88.9%** | +21.9% |
| MiDaS (CNN) | 0.1267 | −3.3% | −5.1% | 512.0 | **+410.1%** | **+865.2%** |

able proxy for geometric consistency under Scrambled Edges. For MiDaS (ViT/CNN), photometric error **decreases** ($-2.3\%$ / $-3.3\%$), suggesting "better" view consistency, yet depth–depth consistency error **increases sharply** ($+89\%$ / $+410\%$), indicating severe geometric inconsistency despite improved appearance-based alignment. This mirrors our single-view metric paradox (Table 4): smooth, boundary-collapsed predictions can reduce pixel-level losses while violating structural constraints. DA V2 shows small changes on these two multi-view metrics, whereas MiDaS exhibits the strongest paradoxical coupling. Energy-matched noise shows different behavior across models (Table 3).

### 4.5. Structural vs. Global Metrics

**(RQ3)** We show that global GT metrics can be misleading under collapse and that boundary-aware structure metrics are required.

**Key finding (metric paradox).** As Table 4 and Appendix E illustrate, global pixel-wise metrics fail to capture these structural failures; instead, edge-alignment metrics are required to expose the loss of boundaries lacking geometric justification.

### 4.6. Extended Validity and Downstream Impact

**(RQ3)** We further validate cross-dataset generalization (KITTI, $\sim 2.1\times$ collapse), task selectivity (SAM segmentation shows no differential sensitivity beyond the high-pass

*Table 4.* **Metric Paradox** ($N=1449$). GT-RMSE (meters, after median-scaling) can *improve* under Scrambled Edges due to smoothing, while structural metrics (Edge F1) reveal severe boundary collapse.

| | GT-RMSE ↓ | | Edge F1 ↑ | | |
|---|---|---|---|---|---|
| Model | Clean | Scram | Clean | Scram | F1 Drop (↓) |
| MiDaS v2.1 (CNN) | 2.19 | **1.45** | 0.195 | 0.105 | 46.4% |
| MiDaS DPT (ViT) | 2.72 | **2.17** | 0.289 | 0.141 | 51.3% |
| DepthAnything v1 (SSL) | 4.29 | **3.53** | 0.364 | 0.187 | 48.8% |
| DepthAnything V2 | 4.08 | **3.51** | 0.392 | 0.131 | **66.7%** |

baseline), long-tail edge cases (reflection/shadow/glass trigger identical collapse patterns), and downstream proxy impacts (surface integrity and traversability metrics show substantial degradation). Full results and methodology are deferred to Appendix B, G, and H. A complete claim-evidence traceability matrix is provided in Appendix F.

## 5. Discussion

### 5.1. Negative Emergence and Occlusion Causality

Our experiments confirm that scaling and SSL pretraining do not, by themselves, yield reliable physical verification behavior (see §1 for the full hypothesis). Among the three priors, violating occlusion causality contributes most strongly to collapse. This aligns with classical depth perception: junction cues (e.g., T-junctions) provide a compact signal for depth ordering (Nakayama & Shimojo, 1992), and corrupting them propagates inconsistent ordering globally, yielding scene-wide structural hallucinations.

### 5.2. Contrast with Human Perception: Anomaly Isolation

The observed failure differs from human visual processing. Humans often treat edges that are physically unsupported (e.g., reflections, spurious shadows) as *local anomalies* rather than evidence requiring a wholesale reinterpretation of global 3D structure—an implicit "detect-and-quarantine" strategy that (i) flags implausible edge cues, (ii) limits their influence on global scene interpretation, and (iii) preserves a coherent 3D hypothesis. Current dense predictors appear to lack such isolation, instead integrating unsupported edges into a globally consistent but incorrect geometry ("Phantom Walls"). This motivates mechanisms for uncertainty estimation, selective cue integration, and graceful degradation under conflict.

### 5.3. Why Normal Estimation Can Be More Sensitive

We observe that surface normal prediction can exhibit stronger collapse than depth (Fig. 5). A natural explanation is that normals are local derivatives of depth ($\mathbf{n} \propto \nabla D$) and are more sensitive to disrupted gradient consistency. Scrambled edges introduce high-frequency discontinuities

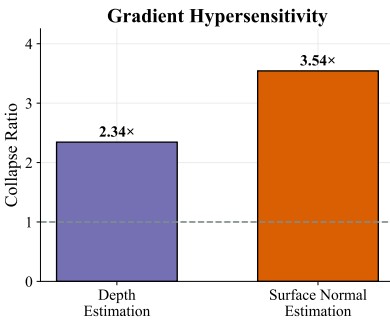

*Figure 5.* **Gradient hypersensitivity.** Surface normal estimation exhibits stronger collapse than depth. Normals are derived from depth via $\nabla D$, showing local derivative consistency degrades more severely.

that disproportionately affect local orientation even when depth becomes globally smoother, explaining why GT pixel-wise metrics may not reflect the severity of structural failure.

### 5.4. Why Global Accuracy Metrics Miss Structural Collapse

Scrambled edges can leave global GT metrics unchanged or even improve them after scale alignment, because models often output smoother predictions that reduce pixel-wise error while losing boundary fidelity (Appendix E). In contrast, boundary-aware metrics reveal the failure: Edge F1 drops substantially (Table 4), indicating corrupted depth discontinuities even when global errors appear small.

### 5.5. Scope and Limitations

We emphasize that Scrambled Edges is a diagnostic perturbation, not a realistic corruption (see §3 for full scope). Our main experiments focus on feed-forward, regression-style dense predictors, and Appendix B.4 extends the check to diffusion- and flow-matching-based depth estimators. These generative models show lower collapse ratios, suggesting partial resilience from delayed or iterative commitment to edge cues, but the effect remains statistically significant. The underlying prior violations arise in real scenarios: reflections, cast shadows, transparency, and motion blur. Our long-tail subset shows elevated sensitivity on such cases (Appendix G), suggesting practical relevance.

The diagnostic intentionally targets edge-to-geometry short-cuts. It does not claim to cover all shortcut channels, such as texture, color shifts, or global illumination changes. This focus is useful because it isolates a specific causal pathway: visually salient edge evidence being converted into geometric structure without sufficient physical support. Tasks that do not rely on edge-to-geometry mapping should not be expected to fail in the same way; our SAM segmentation experiment supports this task selectivity. Conversely, other dense geometric tasks that rely on boundary cues, such as optical flow or stereo matching near motion/occlusion

boundaries, are natural next targets for the same diagnostic framework.

Another limitation is that we do not train new robust predictors. Scrambled Edges could be used as data augmentation, but augmentation alone would mainly encourage invariance to a perturbation pattern; it would not teach the model why one strong edge is a valid physical boundary while another is unsupported. This distinction motivates support-aware objectives or modules that supervise cue selection, rather than only suppressing all strong edge-like inputs.

### 5.6. Cue Selection as the Missing Mechanism

The results point to a cue-selection failure rather than a purely local corruption failure. A dense predictor receives edge evidence $e(x)$, but should condition its use on physical support $s(x)$ from surface continuity, illumination coherence, and occlusion ordering. Geometric Collapse occurs when unsupported but salient edges are adopted into depth discontinuities and then propagated through the model's global scene interpretation. The oracle gap in Fig. 3 makes this timing important: once an unsupported cue has been integrated, post-hoc local repair cannot fully recover the clean prediction because much of the error already lies outside the perturbation mask.

This suggests three concrete mechanism directions. First, models should estimate plausibility or uncertainty for edge cues before integration, rather than only smoothing predictions after collapse. Second, cue fusion should be selective: strong visual gradients should be down-weighted when they conflict with geometric or photometric context, but preserved when they correspond to valid high-curvature boundaries. Third, iterative inference appears useful but incomplete; Marigold and DepthFM reduce collapse severity, yet neither eliminates unsupported-edge adoption. We therefore view iterative refinement as a promising component of physical verification, not a substitute for explicit support-aware cue selection.

A related future experiment is *edge prolongation*: extending an existing scene edge beyond its physically plausible support, rather than relocating and rotating edges. Our mechanism ladder predicts weaker but measurable collapse because the perturbation preserves edge origin and direction while violating surface extent and support. Such milder interventions would help connect the diagnostic to more natural edge ambiguities.

### 5.7. Toward Physical Consistency Checking

Our findings motivate mechanisms that explicitly assess the physical consistency of edge cues before global geometric integration. Promising directions include learned plausibility scoring, uncertainty-aware cue fusion (Ab-

delzad et al., 2019), and objectives that enforce multi-view or inverse-rendering consistency. Hand-crafted filters (e.g., local gradient-consistency heuristics) may partially suppress unsupported cues but risk removing valid high-curvature boundaries; our post-hoc defense experiment in Appendix C.3 confirms this partial-but-limited behavior. Designing learned representations that distinguish invalid from valid edges remains an open challenge.

### 5.8. Implications for Foundation Models in Safety-Critical Use

The "Phantom Walls" induced by unsupported edge cues constitute geometry hallucinations rather than localized noise sensitivity. For safety-critical applications (e.g., robotics, autonomous systems), such global structural failures can be more consequential than modest changes in average pixel-wise error. We therefore advocate evaluating foundation models not only on benchmark accuracy but also on stress tests that probe physical consistency and global error propagation; Scrambled Edges provides a simple probe for this purpose.

## 6. Conclusion

We identify **Geometric Collapse**, a structural failure mode showing that evaluated dense depth predictors can integrate visually salient edge cues without exhibiting behavioral evidence of *physical-causality verification*. Using the *Scrambled Edges* diagnostic—which introduces edge-like cues that violate surface continuity, illumination coherence, and occlusion causality—we observe large increases in prediction deviation relative to energy-matched noise, with effects that propagate globally beyond the perturbed region. Our experiments highlight three takeaways: (i) collapse is driven by prior violations (especially occlusion causality), not edge sparsity or frequency content; (ii) local repairs face a hard ceiling due to global spillover; and (iii) noise robustness does not imply robustness to geometrically unsupported edges. These results motivate architectures that explicitly assess physical-causality and selectively integrate edge evidence.

## Impact Statement

This paper reveals a vulnerability in dense geometric predictors that could affect safety-critical deployments such as robotics, autonomous navigation, and assistive perception. If a model converts unsupported visual edges into global depth structure, downstream systems may infer phantom obstacles, distorted free space, or incorrect surface orientation. The intended positive impact of this work is therefore diagnostic: it provides a controlled test for identifying when dense predictors fail to quarantine physically unsupported

cues.

Scrambled Edges are *not* designed as a practical adversarial attack. They are synthetic interventions used to isolate a failure mechanism under matched controls. The main misuse risk is that such diagnostics could be repurposed to search for deployment failures in safety-critical perception stacks. We mitigate this risk by framing the method as an evaluation protocol and by emphasizing defenses based on plausibility scoring, uncertainty-aware cue fusion, and physically grounded consistency checks.

The computational cost of our protocol is moderate: it mainly requires standard forward passes through existing depth models and lightweight image perturbations. Still, broader benchmark adoption should report compute, model size, and energy usage, especially when evaluating large generative depth estimators. We recommend using the diagnostic selectively as part of robustness audits rather than as an unnecessarily large-scale stress test.

## Acknowledgements

The research presented in this paper was partially supported by the Research Grants Council of the Hong Kong Special Administrative Region, China (CUHK 2300246, RGC C1043-24G), (CUHK 14203425, RGC GRF 2151317), and CUHK 7010870.

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

## A. Related Work Positioning Matrix

Table 5 provides a compact comparison of closely related lines of work, highlighting which components are explicitly isolated in prior diagnostics (e.g., matched controls, physical-prior targeting, and propagation/repair analysis). We include this matrix to clarify how our evaluation protocol and claims differ from robustness-to-noise and geometric-consistency literatures.

## B. Additional Experimental Details

### B.1. Mechanism Ladder: Condition Definitions

Table 6 provides complete definitions for all ablation conditions used in the mechanism ladder (Table 2). Each condition isolates a specific physical prior violation.

**Intensity scale convention.** Throughout the paper we represent RGB images as $I \in [0, 1]$ and apply perturbations in that normalized scale (with clipping to $[0, 1]$ after modification). Values previously written in an 8-bit form (e.g., $180 \cdot \alpha$ and $\pm 150 \cdot \alpha$) should be interpreted as $(180/255) \cdot \alpha$ and $\pm(150/255) \cdot \alpha$ when operating on $I \in [0, 1]$.

**Edge generation details (implementation).** Edge pixels are extracted via Canny (thresholds $\text{thr}_{\text{low}}{=}50$, $\text{thr}_{\text{high}}{=}150$) followed by a $2 \times 2$ morphological dilation. Selected edge components are rendered with a 3-pixel contour thickness; the final perturbation mask is dilated with a $5 \times 5$ kernel (2 iterations). Mean mask coverage is $\approx 36\%$ on NYU.

**Note on T-Junctions.** The T-Junctions condition specifically targets local occlusion ordering by applying random contrast reversals at edge pixels without spatial displacement, simulating scenarios where depth ordering cues at junction boundaries become ambiguous or inverted.

### B.2. Cross-Dataset Validation: KITTI

The dataset contains $N{=}6852$ depth frames with matched RGB images from the KITTI raw split.

### B.3. Detailed Statistics

Table 8 provides the complete statistical breakdown of Geometric Collapse across all evaluated models, identifying significant performance degradation ($p < 0.001$) for every architecture.

### B.4. Cross-Paradigm Generative Depth Estimators

To test whether Geometric Collapse is specific to one-shot feed-forward regressors, we evaluate two generative depth estimators under the same Scrambled-Edges protocol: Marigold (Ke et al., 2024), a diffusion-based estimator, and DepthFM (Gui et al., 2024), a flow-matching estimator.

| Line of work | Edge-level intervention | Matched controls | Targets phy. priors | Measures verify | Prop. / repair ceil. | Boundary eval. |
|---|---|---|---|---|---|---|
| Corruption benchmarks (Hendrycks & Dietterich, 2019) | × | × | × | × | × | × |
| Frequency / shift sensitivity (Yin et al., 2019; Wang et al., 2020; Azulay & Weiss, 2019; Zhang, 2019) | × | × | × | × | × | × |
| Adversarial perturbations / patches (Goodfellow et al., 2015; Brown et al., 2018) | × | × | × | × | × | × |
| MDE robustness / perturbations (Nugent et al., 2025; Chen et al., 2021; Murez et al., 2017) | × | × | × | × | × | × |
| Depth with boundary emphasis (Hu et al., 2018) | × | × | × | × | × | ✓ |
| Geometric consistency / inverse graphics / 3D reps (Gibson, 1979; Marr, 1982; Kersten et al., 2004; Godard et al., 2017; Jaques et al., 2020) | × | × | ✓ | × | × | × |
| Modern monocular depth (scaling / SSL) (Ranftl et al., 2020; 2021; Bhat et al., 2023; Yang et al., 2024; Dosovitskiy et al., 2021; He et al., 2021; Oquab et al., 2024; Kirillov et al., 2023) | × | × | × | × | × | × |
| **Ours: Scrambled Edges / Geometric Collapse** | ✓ | ✓ | ✓ | ✓ | ✓ | ✓ |

*Table 5.* **Positioning matrix.** ✓ indicates the capability is explicitly isolated/quantified as a primary objective in that line of work; × indicates it is not a central target. Our contribution is to test *inference-time physical verification* under controlled edge evidence interventions with matched controls, and to quantify global propagation and its implications for repair and evaluation.

*Table 6.* Complete definitions of mechanism ladder conditions. All conditions use $K=15$ edge segments (Canny: thr$_{\text{low}}$=50, thr$_{\text{high}}$=150) and intensity $\alpha=0.8$ unless noted.

**Baselines (no physical prior violation).**

**High-pass Noise (None)** Gaussian noise $\epsilon \sim \mathcal{N}(0, \sigma_n^2)$ high-pass filtered via $\epsilon - G_\sigma(\epsilon)$; $\sigma_n$ scaled to match RMS energy of Scrambled Edges.

**Edge-Shaped (None)** Noise/color shift applied to original edge locations *in-place* (no translation, no rotation).

**Mask-Matched (None)** Unstructured high-pass noise applied *only* within the Scrambled Edges mask $M_{\text{scram}}$ (same support/position, no edge structure). Since the perturbation is spatially confined, its global RMS energy is lower than global high-pass noise, so its stability ratio can be $< 1$.

**Single-violation conditions.**

**Darkening (Illumination)** Edges darkened by $(180/255) \cdot \alpha$ at original locations (no spatial transform).

**Junction Contrast Reversal (Local Occlusion)** Random brightness $\pm(150/255) \cdot \alpha$ at edge pixels (a proxy for depth-ordering cue disruption at junctions).

**Position (Continuity)** Edges translated by $\pm25\%$ of image dimension; darkened.

**Direction (Causality)** Edges rotated $\pm60°$ around segment centroid; darkened.

**Full violation.**

**Scrambled Edges (All three)** Translation + rotation + darkening combined.

Both use fixed random seeds and the same NYU sample set.

**Interpretation.** The reduced ratios for Marigold and DepthFM suggest that iterative or generative inference can delay or weaken commitment to unsupported edges. However, both models still show statistically significant collapse. This supports a behavioral interpretation: collapse persists when unsupported edge cues are adopted without explicit physical-support verification, although the severity depends on the inference paradigm.

*Table 7.* Cross-dataset validation: Geometric Collapse generalizes to KITTI outdoor scenes ($N$=6852). KITTI collapse ratios are generally lower than NYU, potentially reflecting outdoor scene statistics (larger depth ranges, different textures) and LiDAR ground-truth sparsity.

| Model | NYU Collapse | KITTI Collapse |
|---|---|---|
| MiDaS v2.1 (CNN) | 1.88× | 1.21× |
| MiDaS DPT (ViT) | 2.34× | **2.08×** |
| DepthAnything v1 (SSL) | 2.02× | 1.40× |
| DepthAnything V2 | **3.20×** | 2.48× |

*Table 8.* Statistical significance of collapse ($N$=1449, Paired t-test). All models exhibit highly significant degradation under Scrambled Edges with large effect sizes.

| Model | Noise | Scrambled | Ratio | Cohen's $d$ | $p$-value |
|---|---|---|---|---|---|
| MiDaS v2.1 | 0.130 | 0.244 | 1.88× | 1.37 | $p < 10^{-10}$ |
| MiDaS DPT | 0.077 | 0.181 | 2.34× | 2.11 | $p < 10^{-10}$ |
| DepthAnything v1 | 0.057 | 0.116 | 2.02× | 1.13 | $p < 10^{-10}$ |
| DepthAnything V2 | 0.070 | 0.225 | 3.20× | 2.98 | $p < 10^{-10}$ |

*Table 9.* Cross-paradigm collapse under Scrambled Edges. Generative estimators reduce but do not eliminate the effect.

| Model | Paradigm | Collapse Ratio | Cohen's $d$ | $p$-value |
|---|---|---|---|---|
| MiDaS DPT | Feed-forward ViT | 2.34× | 2.11 | $p < 10^{-10}$ |
| Marigold | Diffusion | 1.55× | 0.88 | $7.38 \times 10^{-139}$ |
| DepthFM | Flow matching | 1.11× | 0.25 | $7.72 \times 10^{-13}$ |

### B.5. Minimal Behavioral Theory of Geometric Collapse

We formalize the failure mode at the behavioral level, without claiming a complete internal mechanistic theory of every dense predictor. Let $M$ denote the perturbation mask, $D_0$ the clean prediction, and $D_1$ the prediction under Scrambled Edges. For each pixel $x$, let $e(x)$ indicate the presence of an edge cue, $s(x)$ its physical support, $a(x)$ whether the model adopts the cue as an aligned depth discontinuity, and $\delta(x) = |D_1(x) - D_0(x)|$ the prediction deviation. In our benchmark, $s(x)$ is operationalized through the geometry, photometric, and occlusion support proxies in Appendix D;

$a(x)$ is operationalized by Adoption Rate; collapse severity is measured by Collapse Ratio; and propagation is measured by out-of-mask error and oracle recovery.

This yields three testable propositions:

1. **Unsupported-edge adoption.** Physically unsupported but visually salient edges can have nonzero adoption probability: $P(a(x) = 1 \mid e(x) = 1, s(x) \approx 0) > 0$.

2. **Collapse monotonicity.** Increasing unsupported-edge adoption should increase collapse severity: $\partial C / \partial \mathbb{E}[a(x)(1 - s(x))] > 0$, where $C$ denotes a collapse measure such as Collapse Ratio.

3. **Spillover repair ceiling.** If perturbation-induced error propagates outside $M$, then any repair operator restricted to $M$ cannot exactly recover $D_0$.

The third proposition gives a strict local-repair bound. Let $R_M$ be any repair operator whose support is restricted to the perturbation mask, so $R_M(D_1)(x) = D_1(x)$ for all $x \notin M$. If there exists any $x \notin M$ such that $D_1(x) \neq D_0(x)$, then $R_M(D_1) \neq D_0$. Thus, nonzero out-of-mask spillover is sufficient to make exact local repair impossible. This is why the oracle analysis is not only diagnostic: it proves that once unsupported cues have propagated globally, mask-local output repair has a hard ceiling.

### B.6. Perturbation Sensitivity Analysis

The sensitivity analyses in Table 10 and Table 10 verify that Geometric Collapse is not an artifact of our default perturbation parameters.

*Table 10.* Sensitivity to $K$ (segments) and $\alpha$ (intensity) ($N$=1449, MiDaS DPT).

| **(a) $K$ sensitivity ($\alpha$=0.8)** | | | | **(b) $\alpha$ sensitivity ($K$=15)** | | |
|---|---|---|---|---|---|---|
| $K$ | Mask Cov. | Collapse | Adopt. | $\alpha$ | Collapse | Adopt. |
| 5 | 27.8% | 2.10× | 37.3% | 0.2 (weak) | 1.37× | 6.2% |
| 10 | 33.1% | 2.27× | 36.8% | 0.4 | 1.62× | 11.5% |
| 15 | 36.0% | 2.34× | 36.1% | 0.6 | 2.02× | 24.7% |
| 20 | 38.0% | 2.39× | 35.5% | 0.8 | 2.34× | 36.1% |

**Definition of Adoption Rate.** We define Adoption Rate as the fraction of scrambled-edge pixels where the predicted depth gradient aligns with the injected edge direction:

$$\text{Adoption Rate} = \frac{\sum_{p \in M_{\text{scram}}} \mathbb{I}(|\nabla D_{\text{pred}}(p)| > \tau_\nabla \wedge \theta(p) < 30°)}{\sum M_{\text{scram}}} \quad (8)$$

where $\tau_\nabla$ is the median gradient magnitude over the image, and $\theta(p)$ is the angle between the predicted depth gradient and the local edge *normal* at pixel $p$ (i.e., the direction perpendicular to the edge tangent). Intuitively, this measures whether the model "adopts" the false edges by creating depth discontinuities across them.

**Interpretation of Sensitivity.** Geometric Collapse emerges systematically as perturbation intensity increases. While weak perturbations ($\alpha = 0.2$) are largely ignored (Adoption Rate 6.2%), collapse becomes pronounced (Ratio $> 2.0\times$) once the model begins to adopt the injected edges (Adoption Rate $> 25\%$ at $\alpha = 0.6$). This confirms that the failure is driven by the *adoption* of false cues, not just their presence.

### B.7. Cross-Task Validation: Semantic Segmentation

We used the Segment Anything Model (SAM) (Kirillov et al., 2023) with ViT-Base backbone in a prompt-free *automatic mask generation* mode (fixed generator settings across all conditions). This choice avoids prompt-selection confounds when comparing clean/noise/scrambled inputs at scale.

**Output representation.** SAM produces a *set* of binary masks rather than a single semantic label map. To obtain a stable, comparable output, we rasterize the mask set into (i) a union-of-masks foreground map $U \in \{0, 1\}^{H \times W}$, and (ii) a boundary map $B \in \{0, 1\}^{H \times W}$ obtained by taking the union boundary of all masks (1-pixel boundaries after morphological edge extraction).

**Pixel Deviation.** For a perturbed condition $c \in \{\text{noise}, \text{scram}\}$, Pixel Deviation is computed against the clean output as

$$\text{PixelDev}(c) = \frac{1}{HW} \sum_p \mathbb{I}(U_c(p) \neq U_{\text{clean}}(p)). \quad (9)$$

**Boundary IoU.** Boundary IoU is computed between boundary maps (again vs. the clean output):

$$\text{BoundaryIoU}(c) = \frac{\sum(B_c \wedge B_{\text{clean}})}{\sum(B_c \vee B_{\text{clean}})}. \quad (10)$$

*Table 11.* Cross-task comparison: Semantic Segmentation ($N$=1449, SAM ViT-Base). Both perturbations cause strong fragmentation; our claim is strictly *differential*: scrambled edges show no additional sensitivity beyond the high-pass baseline.

| Metric | Clean | Noise | Scrambled | Collapse Ratio |
|---|---|---|---|---|
| Pixel Deviation (%) | 0.0 | 67.9 | 57.2 | 0.84× |
| Boundary IoU (%) | 100.0 | 8.67 | 8.67 | 1.00× |

**Interpretation.** The results confirm that scrambled edges do not cause *additional* change beyond the high-pass baseline for this task.

## B.8. Scrambled Edges Algorithm

**Algorithm 1** Scrambled Edges Generation (see Appendix B.1 for details)

**Input:** Image $I$, num_segments $K$, intensity $\alpha$
**Output:** Perturbed image $I_{\text{scram}}$, mask $M_{\text{scram}}$
$E \leftarrow \text{CannyEdgeDetect}(I, \text{thr}_{low} = 50, \text{thr}_{high} = 150)$
$E \leftarrow \text{Dilate}(E, \text{kernel} = 2 \times 2)$
$\{c_1, \dots, c_N\} \leftarrow \text{ConnectedComponents}(E)$
$\{c_1, \dots, c_K\} \leftarrow \text{TopK}(\{c_i\}, \text{by area})$
$M_{\text{scram}} \leftarrow \mathbf{0}_{H \times W}$
**for** $k = 1$ **to** $K$ **do**
$\quad \theta_k \sim \mathcal{U}(-60°, +60°); t_k \sim \mathcal{U}(-0.25W, +0.25W) \times \mathcal{U}(-0.25H, +0.25H)$
$\quad T_k \leftarrow \text{AffineMatrix}(\theta_k, t_k); \quad c'_k \leftarrow \text{WarpAffine}(c_k, T_k)$
$\quad M_{\text{scram}} \leftarrow M_{\text{scram}} \cup c'_k$
**end for**
$M_{\text{scram}} \leftarrow \text{Render}(M_{\text{scram}}, \text{thickness} = 3)$
$M_{\text{scram}} \leftarrow \text{Dilate}(M_{\text{scram}}, \text{kernel} = 5 \times 5, \text{iters} = 2)$
$I_{\text{scram}} \leftarrow \text{clip}(I - \alpha \cdot M_{\text{scram}}, 0, 1)$
**return** $I_{\text{scram}}, M_{\text{scram}}$

## B.9. Model Configurations

Table 12 details the architectural specifications and pretraining datasets for all models used in this study.

*Table 12.* Model configurations used in experiments.

| Model | Backbone | Input Size | Pretrained |
|---|---|---|---|
| MiDaS v2.1 | ResNet-101 | $384 \times 384$ | ImageNet |
| MiDaS DPT | ViT-Large | $384 \times 384$ | ImageNet-21K |
| DepthAnything v1 | ViT-Large | $518 \times 518$ | DINOv2 |
| DepthAnything V2 | ViT-Large | $518 \times 518$ | DINOv2+ |

## B.10. Statistical Tests

All statistical comparisons between conditions on the same dataset (e.g., Scrambled vs. Noise on NYU) use paired t-tests with Bonferroni correction. With 4 models × 6 conditions = 24 comparisons, the corrected significance threshold is $\alpha = 0.05/24 \approx 0.002$. All reported p-values ($< 10^{-10}$) are far below this threshold, ensuring robust conclusions. Cross-dataset comparisons (NYU vs. KITTI) are *descriptive* rather than inferential, as independent sampling precludes paired testing; we report mean collapse ratio differences without p-values.

**Cohen's $d$ Definition:** We compute Cohen's $d$ as a *paired* effect size on the paired differences of the reported metric between the compared conditions: $d = \text{mean}(\Delta)/\text{std}(\Delta)$. Here $\Delta_i$ is defined by the specific comparison in each table/figure (e.g., for NYU stability comparisons $\Delta_i =$

$\text{RMSE}_{\Delta,\text{scram},i} - \text{RMSE}_{\Delta,\text{noise},i}$; for multi-view photometric consistency $\Delta_i = \text{PhotoErr}_{\text{scram},i} - \text{PhotoErr}_{\text{clean},i}$). This paired formulation accounts for per-sample variance.

**Cross-paradigm extension.** Recent monocular depth estimation also includes diffusion-based and flow-matching estimators. Appendix B.4 reports a controlled extension to Marigold and DepthFM with fixed seeds and consistent resizing, showing that collapse is attenuated but not eliminated in these generative paradigms.

## C. Extended Results

### C.1. Multi-View Consistency Extended Analysis

We provide additional details and diagnostics for the multi-view geometric consistency experiment (Section 4.4).

**Cross-Sequence Consistency.** Figure 6 shows that the *change* in out-of-mask photometric error under Scrambled Edges is consistent in direction across all three KITTI Odometry sequences (00, 02, 05). For MiDaS, photometric error *decreases* while depth–depth consistency *increases*, matching the multi-view metric paradox (Table 3).

**Photometric error and validity mask.** We compute photometric reprojection error as mean $\ell_1$ error on RGB in $[0, 1]$ using bilinear sampling. We evaluate only valid projected pixels (in-bounds and positive depth) and, in the main text, report all metrics *outside* the injected perturbation mask $M_{\text{scram}}$.

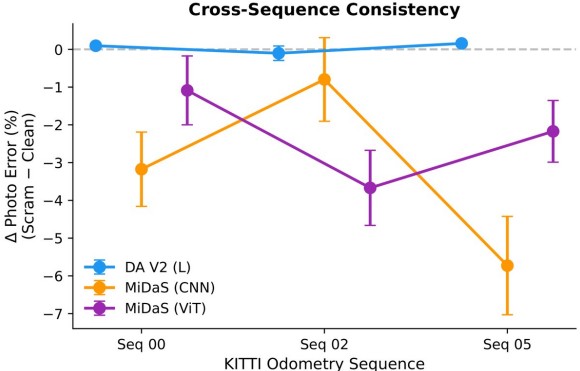

*Figure 6.* **Cross-sequence consistency.** The change in out-of-mask photometric error under Scrambled Edges ($\Delta\%$ vs. Clean) is consistent across all KITTI Odometry sequences. MiDaS shows *decreased* photometric error; Table 3 reports the corresponding depth-consistency increase (multi-view metric paradox). Error bars show 95% CI.

**Depth Consistency Diagnostic.** We also evaluated depth-to-depth consistency (comparing warped vs. target depths). This metric exhibits high variance due to scale ambiguity and heavy-tailed outliers from occlusion boundaries (Figure 7). In the main text, we use *both* photometric and depth consistency to expose the metric paradox: photometric error

can *improve* while depth consistency *degrades*.

## C.2. Detailed Control Experiments

The complete mechanism ladder is presented in Table 2 (Main Text), showing that collapse severity increases with prior violations. Figure 9 visualizes the control conditions.

## C.3. Detailed Defense Analysis

Table 13 details the recovery rates: even with *ground truth* mask knowledge, inpainting ("Input Inpaint (Oracle Mask)") recovers only 24.0% of performance. This stands in stark contrast to the Input Oracle (99.96%), confirming that the geometric collapse induces global error propagation that cannot be fully reversed by local inpainting alone.

*Table 13.* Defense Mechanism Analysis ($N$=1449, MiDaS DPT). Recovery = relative reduction in RMSE$_\Delta$ (vs. clean prediction).

| Method | Level | Recovery |
|---|---|---|
| No Defense | – | 0.0% |
| Mask Coverage | – | 36.0% |
| Outside Error Share | – | **46.7%** |
| **Input Oracle** | **Input** | **99.96%** |
| Input Inpaint (Oracle Mask) | Input | 24.0% |
| **Output Oracle** | **Output** | **47.0%** |
| Output Inpaint (Oracle Mask) | Output | 21.0% |

**Significance of Spillover.** The stark gap between Input Oracle (99.96%) and Output Oracle (47.0%) reveals the fundamental nature of Geometric Collapse: error propagates globally beyond the perturbation mask. Even with perfect mask knowledge, practical inpainting methods achieved limited recovery of 24.0% (input) and 21.0% (output).

**Practical post-hoc defense.** We additionally test lightweight post-hoc defenses on DepthAnything V2. A simple edge-consistency check partially reduces collapse, but remains well below full recovery, consistent with the spillover ceiling.

*Table 14.* Post-hoc defense comparison on DepthAnything V2 ($N$=1449). Recovery is measured relative to the no-defense RMSE vs. clean prediction.

| Defense | RMSE vs. Clean | Recovery | Detection Coverage |
|---|---|---|---|
| No Defense | $0.2252 \pm 0.0650$ | – | 36.0% |
| MSS | $0.1644 \pm 0.0657$ | 23.5% | 0.0% |
| Edge Consistency | $0.1560 \pm 0.0641$ | **26.4%** | 38.7% |
| Combined | $0.1724 \pm 0.0677$ | 24.2% | 10.6% |

These results indicate that explicit plausibility checks are useful, but post-hoc filtering cannot fully reverse errors once unsupported cues have been integrated into the global depth

structure. This motivates support-aware cue selection inside the model rather than purely output-level repair.

## C.4. Ground Truth Edge Alignment

Detailed Edge F1 results are reported in the Main Text. This metric captures *structural* degradation: while global metrics (SSIM, AbsRel) show minimal change due to smoothed predictions, Edge F1 reveals that depth boundary correspondence with ground truth drops significantly.

## D. Additional Proxy Analyses

We provide additional evidence that Scrambled Edges lack not only geometric support but also photometric and occlusion-based justification.

### D.1. Physical Prior Validation

We validate that Scrambled Edges lack physical justification across multiple dimensions:

**Photometric Proxy / P-Score (illumination coherence).** We classify edge pixels by chromatic signature (brightness $\Delta L$ and chromaticity $\Delta C$ in CIE Lab space). Here $L \in [0, 100]$ and $\Delta L, \Delta C$ are measured in Lab units (not in 8-bit RGB intensity). We mark an edge pixel $p$ as *photometrically explained* if it satisfies either a shadow-like signature ($\Delta L(p) > 30$ and $\Delta C(p) < 15$) or a texture/material-like signature ($\Delta C(p) > 15$). We then define

$$\text{P-Score} = \frac{1}{\sum M_{\text{edge}}} \sum_p M_{\text{edge}}(p)\, \mathbb{I}\big(\text{Explained}_{\text{photo}}(p)\big), \tag{11}$$

where $M_{\text{edge}}$ is the edge mask for the corresponding image/condition (Canny + dilation, as specified in Appendix B.1).

**Occlusion Consistency / O-Score (occlusion causality).** We detect candidate T-junctions $\mathcal{J}$ using Harris corners on the edge map, then (i) estimate the local junction orientation by PCA on edge pixels in a neighborhood window, and (ii) test a depth-ordering constraint using the ground-truth depth $D_{\text{gt}}$: the "stem" direction should correspond to the farther surface, while the "bar" corresponds to the nearer occluder. We define

$$\text{O-Score} = \frac{1}{|\mathcal{J}|} \sum_{j \in \mathcal{J}} \mathbb{I}\big(\text{OrderConsistent}(j)\big), \tag{12}$$

where $\text{OrderConsistent}(j)$ is 1 if the measured depth ordering around junction $j$ matches the T-junction occlusion pattern under the estimated orientation.

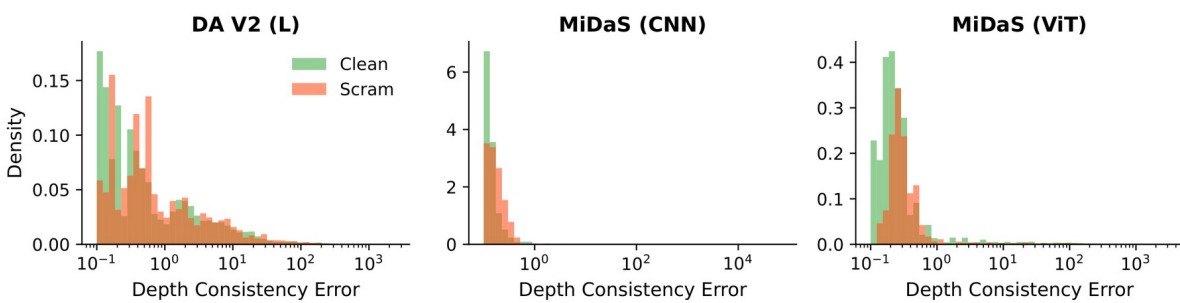

*Figure 7.* **Depth consistency error distribution.** Log-scale histograms show heavy-tailed distributions for depth consistency under all conditions. Photometric reprojection is lower-variance but can be misleading (Table 3); we report both to expose the metric paradox.

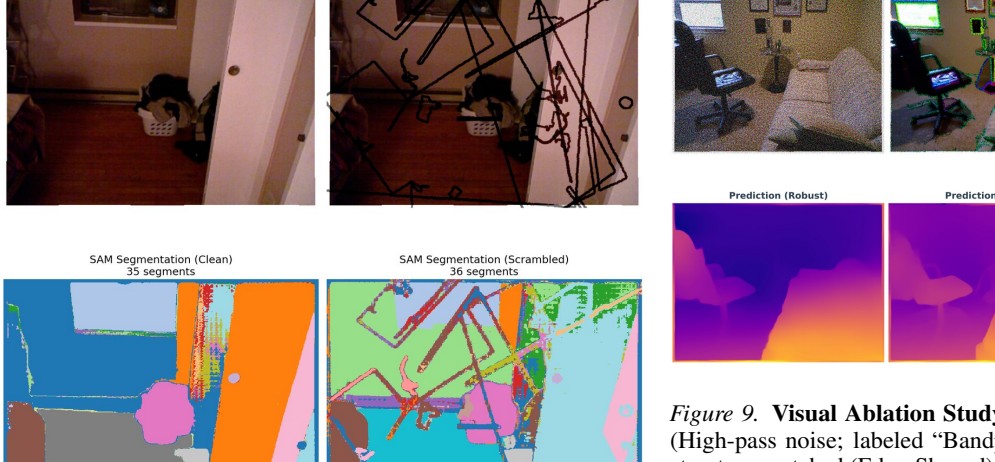

*Figure 8.* **Cross-Task Comparison.** SAM segmentation changes substantially under both perturbations (Table 11). Scrambled edges are not *more* disruptive than high-pass noise (Ratio 0.84×). (SAM ViT-Base, automatic mask generation, top-36 segments.)

*Figure 9.* **Visual Ablation Study.** Comparing energy-matched (High-pass noise; labeled "Bandpass Noise" in the figure) and structure-matched (Edge-Shaped) controls against the Scrambled Edges violation. Note that even with high-intensity Edge-Shaped noise (middle), the model remains robust because geometric causality is preserved. In contrast, Scrambled Edges (right) induce severe collapse.

1. **Smoother predictions align better globally**: Scrambled edges cause the model to produce smoothed depth maps with fewer sharp discontinuities.

2. **Global metrics ignore structural fidelity**: RMSE measures average pixel error, not whether depth boundaries are preserved.

3. **Edge F1 reveals the true failure**: Only structural metrics expose the loss of depth boundaries.

This finding reinforces our central argument: **traditional depth evaluation metrics fail to capture geometric collapse**.

*Table 15.* Physical prior validation ($N{=}1449$). Scrambled Edges lack support across all dimensions.

| Edge Type | Photometric | | Geometry/Occlusion | |
|---|---|---|---|---|
| | Unexplained | Gap | G-Score | O-Score |
| Clean | 23.0% | – | 0.38 | 0.45 |
| Scrambled | **81.5%** | +58.5 | 0.21 | 0.12 |
| Ratio (S/C) | 3.54× | – | 0.55× | **0.27×** |

## E. Why Global Accuracy Metrics are Misleading

As shown in Main Text Table 4, global GT metrics (RMSE, AbsRel) *improve* under Scrambled Edges while Edge F1 degrades by ≈50%. This paradox arises because:

# F. Claim-Evidence Traceability Matrix

To facilitate review and ensure all claims are properly supported, we provide a traceability matrix linking each key finding to its supporting evidence.

*Table 16.* Traceability matrix linking claims to supporting evidence.

| Claim / Key Finding | Evidence | Metric | N |
|---|---|---|---|
| Collapse across models (KF1) | Tab 2, Tab 8 | Collapse ratio | 1449 |
| Not frequency noise (KF2) | Appendix Fig 9 | Ladder collapse | 1449 |
| Sub-additive interaction (KF3) | Fig 9 | Prior ablation | 1449 |
| Spillover ceiling (KF4) | Fig 3 | Recovery | 1449 |
| Behavioral repair bound | Appendix B.5 | Formal implication | – |
| Multi-view metric paradox | Tab 3, Fig 4 | Photo + Depth consist. | 750 |
| Global metrics misleading (KF5) | Tab 4 | Edge F1 | 1449 |
| SSL paradox (KF6) | Tab 2, §4.1 | V2 collapse ratio | 1449 |
| Downstream consequences (KF7) | Tab 17 | Surface var. | 1449 |
| Cross-dataset | Tab 7 | NYU vs KITTI | 6852 |
| Cross-task | Tab 11 | No differential sens. | 1449 |

**Single Source of Truth:** All numerical values in this paper are generated from a unified Results Registry. This ensures consistency across figures, tables, and text, eliminating the risk of conflicting values.

# G. Long-Tail Edge Evaluation Details

## G.1. Case Study Results

To validate that our synthetic perturbation proxies for real-world fragility, we analyze "Long-Tail" edge cases such as reflections (mirrors), shadows, and transparent surfaces (glass). As visualized in Figure 10, these inputs naturally contain edges that violate physical priors (e.g., a reflection creates edges without depth discontinuity). Qualitative analysis confirms that models collapsing under synthetic "Scrambled Edges" exhibit identical artifacts in these real-world scenarios. For instance, mirror reflections are often treated as true depth extensions ("mirror-through" effect), identical to the "transparent wall" hallucinations seen in our synthetic ladder.

The detection criteria for each category are detailed below.

## G.2. Sample Selection Criteria

We automatically classify NYU Depth v2 images into long-tail categories using visual features that detect challenging edge scenarios.

### Reflection Detection:

- Identify high-brightness ($V > 200$), low-saturation ($S < 0.15$) regions in HSV space
- Check for strong RGB edges (Canny) within these regions
- Verify depth flatness (gradient $< 0.02$) in the same regions

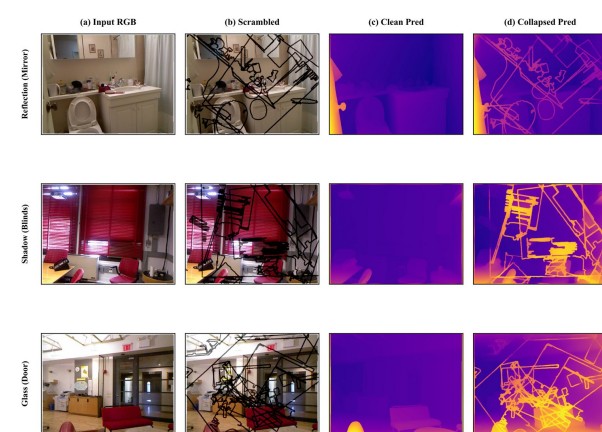

*Figure 10.* **Qualitative Case Study.** Real-world edge ambiguities (Reflections, Shadows, Glass) trigger the same collapse mode as our diagnostic Scrambled Edges, visually confirming the shared failure mechanism.

- Score: proportion of edge pixels in depth-flat, high-brightness regions

### Shadow Detection:

- Compute luminance gradient in CIE Lab space
- Identify strong luminance edges ($|\nabla L| > 30$)
- Check for weak depth gradient ($|\nabla D| < 0.05$) at the same locations
- Score: proportion of luminance edges without depth support

### Glass Detection:

- Identify strong depth gradients ($|\nabla D| > 0.1$)
- Check for weak RGB gradients ($|\nabla I| < 0.2$) at the same locations
- Score: proportion of depth edges without RGB support

### Motion Blur Detection:

- Compute Laplacian variance (classic blur metric)
- Measure edge spreading via dilation ratio
- Score: inverse Laplacian variance $\times$ spreading ratio

**Significance for Real-World Robustness.** The case study demonstrates that images with real-world edge ambiguities (reflection, shadow, glass) are significantly more vulnerable to Scrambled Edges than random samples, validating the practical relevance of our diagnostic perturbation.

# H. Downstream Consequence Evaluation Details

## H.1. Full Downstream Results

Table 17 and Figure 11 below quantify the impact of Geometric Collapse on downstream geometric tasks.

*Table 17.* Complete downstream geometric consequences ($N$=1449, MiDaS DPT). Scrambled Edges induce substantial changes across all geometric proxies.

| Metric | Clean | Scrambled | Change (%) |
|---|---|---|---|
| *Surface Integrity* | | | |
| Normal Variance ↑ | 0.0255 | 0.0577 | +126.6 |
| Planarity Error ↑ | 0.1133 | 0.1552 | +37.0 |
| Folded Surface ↑ | 0.0030 | 0.0049 | +64.9 |
| | | | |
| *Traversability* | | | |
| Fragmentation ↑ | 0.528 | 0.710 | +34.5 |
| Navigable Area ↑ | 0.014 | 0.022 | +57.3 |

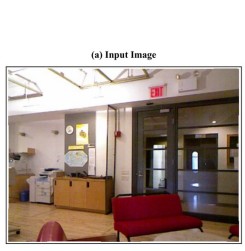
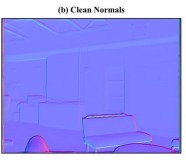
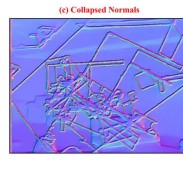
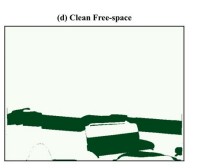
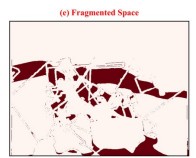

(a) Input Image  (b) Clean Normals  (c) Collapsed Normals  (d) Clean Free-space  (e) Fragmented Space

*Figure 11.* (MiDaS DPT, NYU Depth v2 sample #35) **Downstream Consequences.** Visualizing the impact of Geometric Collapse on surface normal estimation (top) and free-space fragmentation (bottom). Note the massive increase in surface normal variance (noisy, incoherent directions) in the collapsed prediction.

## H.2. Surface Integrity Metrics

**Normal Variance:** For each $5 \times 5$ patch, we compute surface normals from depth gradients and measure the variance of normal vectors within the patch. High variance indicates inconsistent surface orientation.

**Planarity Error:** For each $7 \times 7$ patch, we fit a plane using least squares and compute the RMS residual. High error indicates non-planar local structure.

**Folded Surface Ratio:** We detect potential surface folding by identifying pixels where both curvature magnitude and gradient direction change exceed thresholds. This indicates geometrically impossible self-intersecting surfaces.

## H.3. Traversability Metrics

**Ground Plane Estimation:** We fit a plane to the bottom 30% of the depth map (assumed to be ground) using RANSAC-style robust fitting.

**Free-Space Fragmentation:** We identify navigable regions (within 0.1 of ground plane) and compute connected components:

$$\text{Fragmentation} = 1 - \frac{\text{largest component}}{\text{total navigable area}}. \quad (13)$$

**Navigable Area Ratio:** Proportion of image pixels within navigable height threshold of the estimated ground plane.

## H.4. Metric Definitions

# I. Reproducibility Checklist

## I.1. Code and Data

**Code**: Code and benchmark will be released publicly upon publication. Implementation uses PyTorch 2.0+ with standard libraries (OpenCV, NumPy, SciPy).

**Dataset**: NYU Depth v2 labeled set (Silberman et al., 2012) containing $N$=1449 RGB-D pairs ($480 \times 640$ resolution). Available at https://cs.nyu.edu/~silberman/datasets/nyu_depth_v2.html.

## I.2. Experimental Settings

*Table 18.* Hyperparameters and experimental settings.

| Parameter | Value |
|---|---|
| *Reproducibility* | |
| Random seed | 42 |
| Per-sample seed | $42 + i$ (where $i$ is sample index) |
| *Scrambled Edges parameters* | |
| Number of segments ($K$) | 15 |
| Edge intensity ($\alpha$) | 0.8 |
| Canny thresholds | (50, 150) |
| Rotation range | $\pm 60°$ |
| Translation range | $\pm 25\%$ W/H |
| *Sample size (all experiments)* | |
| NYU Depth v2 | $N$=1449 (full labeled set) |

**Note on Sample Size:** All experiments use the complete NYU Depth v2 labeled set ($N$=1449) to ensure statistical robustness and reproducibility. We do not subsample for any experiment.

## I.3. Computational Resources

- **Hardware**: NVIDIA RTX 4090 (24GB VRAM)

- **Inference precision**: FP16 mixed precision

- **Inference speed**: ∼5 images/second per model

- **Total runtime**: ∼1.5 hours for complete experiment suite

### I.4. Model Versions

- MiDaS v2.1 (CNN): `intel-isl/MiDaS:MiDaS` (ResNet-101)

- MiDaS DPT (ViT): `intel-isl/MiDaS:DPT_Large` (ViT-Large)

- DepthAnything v1: `LiheYoung/depth-anything-large-hf` (ViT-Large, DINOv2)

- DepthAnything V2: `depth-anything/Depth-Anything-V2-Large` (ViT-Large, DINOv2+)

**Inference Details:**

- **Input preprocessing**: Images resized to model-native resolution (MiDaS: $384 \times 384$; DepthAnything: $518 \times 518$) using bilinear interpolation, maintaining aspect ratio with center padding.

- **Normalization**: ImageNet mean/std for MiDaS; model-specific normalization for DepthAnything.

- **Model output**: Inverse relative depth (up to an unknown global scale).

- **Stability RMSE ($RMSE_\Delta$)**: for behavioral stability metrics (e.g., Collapse Ratio), outputs are rescaled to $[0, 1]$ per image (after resizing back to the input resolution) so that deviations are compared in a normalized space.

- **GT-RMSE**: for Table 4, predictions are resized to the NYU resolution and aligned to metric depth using *median scaling* (scale $= \mathrm{median}(D_{gt})/\mathrm{median}(D_{pred})$ on valid pixels), then RMSE is computed in the original depth units (meters).

### I.5. Statistical Tests

All statistical comparisons use two-tailed paired t-tests with Bonferroni correction for multiple comparisons. Effect sizes are reported as Cohen's $d$ (paired formulation).

**Effect Size Interpretation:** Following standard conventions, we interpret $d < 0.5$ as small, $0.5 \leq d < 0.8$ as moderate, and $d \geq 0.8$ as large. All main experiments yield $p < 10^{-10}$ with effect sizes in the moderate to very large range, confirming that Geometric Collapse is a robust and statistically significant phenomenon across all tested models (see Appendix Table 8 for detailed statistics and Appendix Section C for summary results).

