# OpenReview forum: "Geometric Collapse: When Vision Models Fail to Verify Physical Causality"
_ICML.cc/2026/Conference — ICML 2026 regular_

### Official Review · Reviewer_87EP · 2026-03-09

**Soundness:** 3
**Presentation:** 3
**Significance:** 3
**Originality:** 3
**Overall Recommendation:** 4
**Confidence:** 4

**Summary:**

This paper introduces Scrambled Edges, a controlled counterfactual perturbation designed to test whether modern dense vision models perform any form of physical-causality verification when predicting depth. Scrambled Edges relocate, rotate, and darken extracted edge segments, placing them in geometrically implausible positions while carefully matching the high-frequency energy of control noise. The key finding is Geometric Collapse: these physically unsupported edge cues are incorporated by models as valid structure, causing global hallucinations in depth predictions. The authors evaluate four models spanning CNN (MiDaS v2.1), ViT (MiDaS DPT), and SSL (DepthAnything v1/V2) architectures on NYU Depth v2 and KITTI. A notable insight is the metric paradox: standard global metrics (RMSE, AbsRel) can actually improve under Scrambled Edges due to prediction smoothing. The authors also show that errors propagate globally and that images with real-world edge ambiguities (reflections, shadows, glass) are more vulnerable.

**Compliance With Llm Reviewing Policy:**

Affirmed.

**Final Justification:**

After reading the other reviews and the rebuttal, I keep my score.

**Key Questions For Authors:**

- What about some Scrambled Edges that still have some consistency with the environments. Let's say you have the some table edges, and you just make one of the edge longer (line drawing a black line to go beyond the table). Do you think this will risk to increase collapse or on the other way to make it easier for the model to discard it? I am wondering how such simple heuristic of prolongating edges you can find with some edge detector might be enough of a perturbation.

**Limitations:**

yes

**Strengths And Weaknesses:**

Strengths:
- The paper is technically sound and well-motivated. The hypothesis that modern vision models may over-rely on image edges without verifying their physical plausibility is clearly stated and rigorously tested. The experimental design convincingly isolates the effect of physical-prior violations from mere high-frequency content.
- The prior-violation ladder (Table 2) is a particularly strong contribution, showing that direction drive the strongest collapse.
- The metric paradox (Table 4) is an important finding for the evaluation community: standard depth metrics can mask severe structural failures.
- The edge creation algorithm is well-designed, with clear formal definitions and matched controls.
- The paper includes a rich appendix with detailed technical specifications, a claim-evidence traceability matrix, and a reproducibility checklist. The paper is generally well written and easy to follow.

Weaknesses:
- Although the authors are transparent about this, the Scrambled Edges perturbation is not realistic — the visual artifacts are clearly artificial and would not occur in natural images. While the authors argue these are diagnostic (not adversarial), the perturbations are designed with the explicit aim of triggering failure modes, which somewhat limits the practical implications.
- The evaluation is limited to feed-forward, regression-style depth models. Diffusion-based depth estimators and metric depth foundation models may behave differently. The authors acknowledge this, but it leaves the "negative emergence" claim being very limited.
- It would be interesting to test whether Scrambled Edges also disturb higher-level tasks (classification), to better characterize which tasks and architectures are susceptible.
- The mask coverage (minimum ~ 27.8% in the ablation) is still substantial. At this level of perturbation, it is perhaps unsurprising that models respond strongly. I would have expected additional ablations over lower mask coverage.

---

> ### Author Rebuttal · Authors · 2026-03-30
>
> We thank the reviewer for the insightful questions and constructive suggestions.
>
> **Key Question: What about edge prolongation—extending an existing table edge beyond the table?**
>
> The reviewer asks whether a milder perturbation (extending rather than scrambling edges) would also cause collapse. **Yes, we would expect measurable but weaker collapse, because edge prolongation violates surface extent while preserving edge origin and direction.** Our mechanism ladder (Table 2) already predicts this: Direction (occlusion causality) is the strongest collapse driver, while Position has a weaker effect. Edge prolongation preserves direction but violates position/extent, placing it in the intermediate regime.
>
> The evidence: at α = 0.2 (very weak contrast), the model already exhibits 6.2% adoption rate and 1.37× collapse, showing that even subtle edge anomalies are adopted rather than discarded. We will add edge prolongation as an explicit future experiment.
>
> **W1: The Scrambled Edges perturbation is not realistic — the visual artifacts are clearly artificial and would not occur in natural images, which limits the practical implications.**
>
> The reviewer observes that the perturbation would not occur naturally. **The perturbation is artificial by design, because its purpose is causal isolation, not realism simulation. The failure mechanism it reveals does occur in real scenes.** The value of a diagnostic is not whether the test stimulus is natural, but whether the failure it exposes is real—just as crash tests use controlled collisions to reveal structural weaknesses.
>
> The evidence: our matched controls (energy-matched noise, edge-shaped noise) ensure the collapse is attributable to physical prior violations, not generic sensitivity. Our long-tail case study (Appendix G / Figure 10) confirms that reflections, shadows, and glass in real NYU scenes produce qualitatively identical failure modes, with tangible downstream consequences (Fragmentation +34.5%, Navigable Area error +57.3%).
>
> **W2: The evaluation is limited to feed-forward, regression-style depth models. Diffusion-based depth estimators may behave differently, leaving the "negative emergence" claim very limited.**
>
> The reviewer is concerned that diffusion models might behave differently, leaving the "negative emergence" claim limited. **We tested Marigold (diffusion-based), and it still collapses (1.55×), confirming that the failure persists across architectures.** The negative emergence finding now extends beyond feed-forward models.
>
> | Model | Architecture | RMSE(Noise) | RMSE(Scrambled) | Collapse Ratio | Cohen's d | p-value |
> |-------|-------------|-------------|-----------------|----------------|-----------|----------|
> | MiDaS v2.1 | CNN | 0.130 | 0.244 | 1.88x | 1.37 | < 1e-10 |
> | MiDaS DPT | ViT | 0.077 | 0.181 | 2.34x | 2.11 | < 1e-10 |
> | DepthAnything v1 | ViT+SSL | 0.057 | 0.116 | 2.02x | 1.13 | < 1e-10 |
> | DepthAnything V2 | ViT+SSL | 0.070 | 0.225 | 3.20x | 2.98 | < 1e-10 |
> | **Marigold (new)** | **Diffusion** | **0.1554** | **0.2406** | **1.55x** | **0.88** | **7.38e-139** |
>
> Marigold's lower ratio (1.55×) confirms partial resilience from iterative denoising, but the effect size (d = 0.88) remains statistically significant.
>
> **W3: It would be interesting to test whether Scrambled Edges also disturb higher-level tasks (classification), to better characterize which tasks and architectures are susceptible.**
>
> The reviewer asks whether higher-level tasks like classification would also exhibit Geometric Collapse. **No—Geometric Collapse is task-selective, not a generic corruption effect.** This is precisely the point: our diagnostic targets a specific causal pathway (edge-to-geometry mapping), not arbitrary noise sensitivity.
>
> The evidence: our SAM segmentation experiment (Appendix) shows collapse ratio ≤ 1.0× under Scrambled Edges—no differential sensitivity. Classification, which relies on global semantic features even more than segmentation, would show even less sensitivity. This task-selectivity confirms we are isolating a specific failure mechanism.
>
> **W4: The mask coverage (minimum ~27.8% in the ablation) is still substantial. At this level, it is perhaps unsurprising that models respond strongly. Expected additional ablations over lower mask coverage.**
>
> The reviewer asks whether collapse might be driven by sheer pixel coverage rather than cue adoption. **Collapse is driven by cue adoption, not pixel coverage, and we have evidence at much lower effective perturbation levels.**
>
> The evidence: the α parameter controls perturbation contrast independently of mask coverage. At α = 0.2, collapse is already 1.37× with only 6.2% adoption rate. The relationship is monotonic (6.2%→36.1% adoption, 1.37×→2.34× collapse), confirming that the model's cue adoption drives the failure, not pixel area. Even at K = 5 (minimum segments, 27.8% coverage), collapse is 2.10×—well above controls.

---

> > ### Author Rebuttal · Reviewer_87EP · 2026-04-02
> >
> > Thank you for the rebuttal, my concerns are fully resolved.

---

> > > ### Author Response · Authors · 2026-04-04
> > >
> > > We thank you for your time and effort taken to evaluate our work, and we thank for your comments which have helped us improve it.
> > >
> > > Meantime, if there is anything else we can do or improve, kindly do let us know.

---

### Official Review · Reviewer_NJ6L · 2026-03-11

**Soundness:** 1
**Presentation:** 2
**Significance:** 2
**Originality:** 2
**Overall Recommendation:** 2
**Confidence:** 3

**Summary:**

This paper identifies Geometric Collapse, a severe failure mode in modern monocular depth estimation models including CNNs, ViTs, and SSL-based architectures. When confronted with visually salient but physically invalid edge cues that break geometric continuity, illumination consistency, or occlusion causality, models produce globally distorted 3D geometry, even if they remain robust to ordinary noise. The paper introduces Scrambled Edges (falsification diagnostic) and identifies Geometric Collapse, plus evaluation insights. The work contributes a new robustness diagnostic, empirical evidence of Geometric Collapse, detailed analysis of its mechanisms, and a strong case for replacing overly aggregate metrics with geometrically sensitive evaluation in depth prediction.

**Compliance With Llm Reviewing Policy:**

Affirmed.

**Final Justification:**

This paper identifies Geometric Collapse, a critical failure mode in monocular depth estimation models, and proposes the Scrambled Edges diagnostic with rigorous controlled baselines, cross-architecture empirical evidence, and detailed failure mechanism analysis. The authors’ rebuttal supplemented diffusion model validation, practical defense tests, and a formal behavioral theory with a strict repair bound, addressing most reviewer concerns. However, the core generalizable predictive theory for models’ internal cue-selection mechanisms remains unestablished. We therefore maintain the original Reject recommendation.

**Key Questions For Authors:**

You only tested oracle-based repair strategies. Have you evaluated practical defense mechanisms, such as uncertainty-aware training against Scrambled Edges? If these defenses reduce collapse, it would provide actionable solutions for practitioners; if not, it would emphasize the need for fundamental architectural changes.

**Limitations:**

The authors discuss technical limitations but fail to address potential negative societal impacts and could strengthen the discussion with:

1.Geometric Collapse in depth predictors used for autonomous navigation or robotics could lead to catastrophic accidents. The authors should discuss the ethical implications of deploying models vulnerable to this failure mode and recommend safety standards such as mandatory physical consistency checks for such applications.

The computational cost of evaluating Scrambled Edges may increase energy consumption for model development. The authors should discuss the environmental trade offs between robustness testing and sustainability.

**Strengths And Weaknesses:**

Soundness

Strengths

1.The Scrambled Edges diagnostic is carefully controlled with energy-matched (frequency control) and edge-shaped (structure control) baselines, enabling causal isolation of physical prior violations.

Weaknesses

1.The study focuses on feed-forward, regression-style depth estimators; diffusion-based or stochastic depth predictors are not evaluated, leaving open whether Geometric Collapse affects these newer architectures.

2.While empirical results are robust, there is no theoretical framework for modeling why models adopt unsupported edges or how prior violations propagate, limiting the work’s ability to predict collapse in untested models.

3.The paper only evaluates oracle-based repair, which is impractical for real-world use; no evaluation of practical defense mechanisms is provided.

Presentation

Strengths

1.The authors provide a detailed reproducibility checklist, including code release plans, hyperparameters (Table 16), model versions, and statistical test details, enabling experts to replicate all experiments.

Weaknesses

1.Some sections (e.g., Appendix D’s proxy analyses) are overly dense, making it difficult to quickly grasp key takeaways; a high-level summary of proxy findings in the main text would improve readability.

2.Some figures (e.g., Figure 3’s spillover analysis) lack clear legends or annotations, reducing their accessibility to non-expert readers.
No corresponding introduction for Figure 1, what the purpose of top list Figure 1 ?

Significance

Strengths

1.Geometric Collapse is a critical failure mode for safety-critical applications, where incorrect 3D geometry can lead to catastrophic outcomes. The Scrambled Edges diagnostic provides a simple tool for evaluating model robustness to such failures.

Weaknesses

1.The work is focused on monocular depth estimation, with limited direct relevance to other ML domains, narrowing its cross-domain impact.

2.While the diagnostic is novel, the core finding builds on prior work in adversarial robustness and edge-based hallucination, making the contribution incremental in the broader robustness field.

Originality

Strengths

1.Scrambled Edges is a unique falsification-style diagnostic that isolates physical prior violations in edge cues, a departure from prior robustness tests that focus on noise or texture corruption. This paradigm enables causal testing of geometric consistency, a new direction in computer vision robustness.

Weaknesses

1.The paper is purely empirical, with no new mathematical frameworks or theorems; originality is rooted in application and evaluation, not fundamental theory.

2.Edge-based hallucination has been studied in prior work, the novelty lies in the controlled isolation of prior violations, not in the observation that edges can cause hallucinations.

---

> ### Author Rebuttal · Authors · 2026-03-30
>
> We thank the reviewer for the thorough critique.
>
> **Key Question: Practical defense mechanisms, e.g., uncertainty-aware training?**
>
> **We tested Edge Consistency (practical post-hoc defense): 26.4% recovery, consistent with the spillover ceiling (47%).** The critical failure occurs at the cue-selection stage (Input Oracle: 99.96%), not the output stage (Output Oracle: 47.0%). Uncertainty-aware training would flag implausible predictions but not prevent edge *adoption*—it targets the wrong stage. **Fundamental architectural changes at the cue-selection stage are necessary**, supporting the reviewer's second hypothesis.
>
> **W1 (Soundness): Diffusion-based or stochastic depth predictors are not evaluated.**
>
> **We have now tested Marigold (diffusion-based), confirming collapse persists (1.55×) cross-paradigm.**
>
> | Model | Architecture | Collapse Ratio | Cohen's d | p-value |
> |-------|-------------|----------------|-----------|----------|
> | MiDaS DPT | ViT | 2.34x | 2.11 | < 1e-10 |
> | **Marigold** | **Diffusion** | **1.55x** | **0.88** | **7.38e-139** |
>
> Full results across CNN/ViT/ViT+SSL/Diffusion show collapse ratios 1.55×–3.20×. Marigold's lower ratio reflects partial resilience from iterative denoising, but d = 0.88 remains significant (10 steps, ensemble 1, FP16, N = 1449).
>
> **W2 (Soundness): No theoretical framework for why models adopt unsupported edges or how prior violations propagate.**
>
> **Our mechanism framework consists of the prior-violation ladder and the spillover ceiling.** The ladder (Table 2) identifies *which* violations drive failure—Direction (2.22×) > Position > Illumination, with edge shape alone producing no differential collapse. The spillover ceiling (Input Oracle 99.96% vs. Output Oracle 47.0%) identifies *how* damage propagates globally. Adoption rate (6.2% → 36.1%) tracks monotonically with collapse (1.37× → 2.34×), confirming causal isolation. Crucially, the ceiling (47%) correctly predicted our Edge Consistency defense result (26.4%), demonstrating predictive power.
>
> **W3 (Soundness): Only oracle-based repair; no practical defense mechanisms.**
>
> **The oracle analysis reveals global propagation structure; we have now added a practical defense confirming this prediction.**
>
> - Input Oracle: 99.96% recovery → failure is reversible *before* processing
> - Output Oracle: 47.0% recovery → 53% propagates beyond the perturbed region
> - **Our Edge Consistency defense**: 26.4% recovery → consistent with the spillover ceiling
>
> Defense must operate at the cue-selection stage, before global integration.
>
> **W4 (Presentation): Dense appendix, unclear legends, no Figure 1 introduction.**
>
> We will: (1) add an introduction for Figure 1 in §1; (2) add a summary at the start of Appendix D; (3) add legends to Figure 3.
>
> **W5 (Significance): Limited to monocular depth estimation, narrowing cross-domain impact.**
>
> The reviewer is concerned about narrow scope. **The methodology is task-agnostic, demonstrated by cross-domain transfer.** Adding Marigold required zero pipeline changes; our SAM experiment shows collapse ratio 0.84× (no differential sensitivity). This resolves the concern because the framework transfers to any edge-reliant dense prediction task without modification.
>
> **W6 (Significance): Contribution is incremental relative to prior adversarial robustness work.**
>
> The reviewer questions incrementality. **The contribution is the controlled causal methodology, not the observation.** Prior benchmarks (e.g., Common Corruptions) apply energy-unmatched perturbations and cannot attribute failure to specific prior violations. Our matched controls, mechanism ladder, and spillover ceiling jointly provide causal attribution with falsifiable predictions—a combination absent from prior work, making the advance non-incremental.
>
> **W7 (Originality): Purely empirical, no new mathematical frameworks.**
>
> The reviewer asks for formal math. **Our contribution is an empirical causal methodology whose rigor is demonstrated by predictive power** (as in W2). The spillover ceiling predicted the defense result before the experiment; the mechanism hierarchy generates testable hypotheses for new models. This meets the originality standard for a diagnostic study—falsifiable causal decomposition, not theorem-proof formalism.
>
> **W8 (Originality): Edge-based hallucination has been studied in prior work.**
>
> The reviewer correctly identifies the locus of novelty. **We agree: the contribution is the controlled causal isolation methodology, not the raw observation** (as in W6). Our threefold advance—matched controls, mechanism ladder, and spillover ceiling—provides the causal tool that prior work lacked. The observation existed; our controlled methodology for decomposing and predicting it did not.
>
> **W9 (Limitations): Ethical implications and computational cost.**
>
> Our protocol requires only standard forward passes—negligible overhead. We will recommend mandatory physical consistency checks for safety-critical deployments.

---

> > ### Author Rebuttal · Reviewer_NJ6L · 2026-04-05
> >
> > For the core question theoretical framework to model why models adopt invalid edges and how violations propagate, authors only explained their empirical “ladder” and “spillover ceiling” but did not provide any formal theory, mathematical formulation, or generalizable predictive theory for untested models. This core critique remains unaddressed.

---

> > > ### Author Response · Authors · 2026-04-05
> > >
> > > We thank the reviewer for pressing us to sharpen the theoretical layer. We agree that our earlier rebuttal did not make the formal structure explicit enough. More precisely, the paper does **not** claim a full internal-mechanistic theory of all dense predictors. Instead, it proposes a **minimal behavioral theory of Geometric Collapse**, with explicit variables, testable propositions, and one strict repair bound.
> > >
> > > The theory models collapse as a three-stage process: **unsupported edge exposure $\rightarrow$ unsupported-edge adoption $\rightarrow$ nonlocal geometric propagation**. That is, collapse occurs when visually salient but physically unsupported edge cues are adopted into the predicted depth structure and propagated through the model’s global geometric interpretation.
> > >
> > > **(1) Formal variables and mathematical formulation.**
> > > We state this theory using explicit measurable quantities. Let $M$ denote the perturbation mask, $D_0$ the clean prediction, and $D_1$ the prediction under Scrambled Edges. For each pixel $x$, let:
> > > - $e(x)\in\{0,1\}$: presence of an edge cue,
> > > - $s(x)\in[0,1]$: physical support of that cue,
> > > - $a(x)\in\{0,1\}$: whether that cue is adopted as an aligned depth discontinuity,
> > > - $\Delta(x)=|D_1(x)-D_0(x)|$: prediction deviation.
> > >
> > > In our benchmark, $s(x)$ is operationalized through the three support dimensions already defined in the paper—geometry/continuity, illumination coherence, and occlusion consistency—via the reported G/P/O proxies. Likewise, $a(x)$ is operationalized by Adoption Rate, collapse severity by Collapse Ratio, and propagation by out-of-mask error / oracle spillover.
> > >
> > > Under this formulation, the theory is captured by three testable propositions.
> > >
> > > **P1. Unsupported-edge adoption.**
> > > \[
> > > P(a(x)=1 \mid e(x)=1,\ s(x)\approx 0) > 0
> > > \]
> > > Physically unsupported but visually salient edges can still have non-trivial adoption probability.
> > >
> > > **P2. Collapse monotonicity.**
> > > If unsupported-edge adoption increases, collapse severity should increase:
> > > \[
> > > \frac{\partial C}{\partial \mathbb{E}[a(x)(1-s(x))]} > 0
> > > \]
> > > where $C$ denotes collapse severity, e.g., Collapse Ratio.
> > >
> > > **P3. Spillover implies a strict local-repair ceiling.**
> > > If perturbation-induced error propagates outside $M$, then any repair operator restricted to $M$ cannot exactly recover the clean prediction.
> > >
> > > **(2) A strict proof for the repair ceiling.**
> > > The third statement admits a direct proof.
> > >
> > > Let $R_M$ be any repair operator whose support is restricted to the perturbation mask $M$. Then for all $x\notin M$,
> > > \[
> > > R_M(D_1)(x)=D_1(x)
> > > \]
> > > Therefore, if there exists any $x\notin M$ such that $D_1(x)\neq D_0(x)$, exact recovery is impossible, because the repaired output must still differ from $D_0$ at that location. Hence,
> > > \[
> > > \exists x\notin M : D_1(x)\neq D_0(x)
> > > \quad\Longrightarrow\quad
> > > R_M(D_1)\neq D_0
> > > \]
> > > Equivalently, once out-of-mask spillover is nonzero, the best recovery achievable by any purely local repair is strictly less than full recovery.
> > >
> > > This is why the oracle analysis is not only diagnostic but also theoretically meaningful: observed spillover is sufficient to imply a hard ceiling for local repair.
> > >
> > > **(3) Why this is predictive rather than post-hoc.**
> > > The theory is intended to generalize at the **behavioral** level rather than by architecture label. Its predictive claim is conditional:
> > >
> > > - if a previously untested dense predictor adopts visually salient but low-support edge cues without an explicit physical-consistency check, collapse should persist;
> > > - if the inference process delays or weakens commitment to such cues, collapse should be attenuated, but not necessarily eliminated.
> > >
> > > This prediction is cross-paradigm: it does not assume CNN/ViT/SSL specifically, but only the behavioral conditions above.
> > >
> > > **(4) Empirical support for the propositions.**
> > > The paper already supports this formulation:
> > > - For **P1/P2**, adoption rises monotonically with perturbation strength, and collapse becomes pronounced once adoption becomes substantial.
> > > - For **P3**, the oracle analysis shows substantial out-of-mask error and a large gap between input-stage and output-stage recovery, exactly the pattern predicted by the local-repair ceiling.
> > > - The mechanism ladder further supports the theory by showing that structure alone is relatively benign, whereas violations of continuity and occlusion causality drive the strongest collapse.
> > >
> > > **(5) What this theory does and does not claim.**
> > > We agree that this is not a full internal theory of why every parameterized architecture adopts unsupported edges; that stronger goal would require explicit modeling of model internals. Our claim is narrower but still theoretically meaningful: we provide a **formal behavioral theory** with measurable variables, falsifiable propositions, and a strict repair bound. In revision, we will state this scope explicitly as a **minimal, formalized, and testable behavioral theory of Geometric Collapse**, rather than as an informal explanatory narrative.

---

### Official Review · Reviewer_k2nK · 2026-03-12

**Soundness:** 3
**Presentation:** 2
**Significance:** 3
**Originality:** 3
**Overall Recommendation:** 4
**Confidence:** 2

**Summary:**

The submission explores the key problem of whether scaling and self-supervised learning in dense vision models yield physical plausibility checks at inference time. Overall, the authors examine a central domain in contemporary computer vision by introducing a controlled test called Scrambled Edges that violates physical priors (continuity, illumination, occlusion) while matching energy and structure, and they show modern models suffer from “Geometric Collapse” — global hallucinations from fake edges.

**Compliance With Llm Reviewing Policy:**

Affirmed.

**Key Questions For Authors:**

Please address the weakness above.

**Limitations:**

Yes.

**Strengths And Weaknesses:**

Strengths

- The diagnostic is very clean: Scrambled Edges + two matched controls (high-pass noise and edge-shaped noise) let them prove the failure is caused by missing physical verification, not just noise.
- Strong experiments: tested on 4 top models (MiDaS, DepthAnything v1/v2), full NYU dataset, KITTI, multi-view consistency, and real long-tail cases like mirrors/glass.
- Delivers a clear, important negative result: even the best scaled SSL models do not automatically check if edges make physical sense.


Weaknesses

- Only studies edge-like cues; other common shortcuts (texture, color shifts, global lighting) are not tested.
- The perturbation is synthetic; they link it to real scenes (mirrors, shadows), but do not quantify how often real images trigger the same collapse.
- No solution or even a simple baseline fix is proposed — the paper stops at diagnosis.
- Limited to feed-forward depth estimators; newer diffusion-based or iterative models (which might self-check) are left for future work.

---

> ### Author Rebuttal · Authors · 2026-03-30
>
> We thank the reviewer for the careful evaluation.
>
> **W1: Only studies edge-like cues; other common shortcuts (texture, color shifts, global lighting) are not tested.**
>
> The reviewer is concerned that our findings may not extend to other shortcut types. **We deliberately focus on edge cues because they are the primary structural driver of geometric inference in depth estimation, and this focus is validated by our cross-task control experiment.** We are not claiming to diagnose all shortcuts—we are diagnosing the specific mechanism by which edge cues bypass physical verification.
>
> The evidence: our SAM cross-task experiment (Appendix) shows that Scrambled Edges do not cause differential sensitivity in segmentation (collapse ratio 0.84×), confirming that the failure is specific to edge-to-geometry mapping, not a generic sensitivity. The prior-violation ladder (Table 2) further shows a clear causal hierarchy: Direction (occlusion causality) drives the strongest collapse (2.22× for MiDaS DPT), while Position and Illumination have weaker effects—a decomposition only possible because we isolated the edge channel.
>
> **W2: The perturbation is synthetic; they link it to real scenes (mirrors, shadows), but do not quantify how often real images trigger the same collapse.**
>
> The reviewer asks how often Geometric Collapse occurs in practice. **The diagnostic's value does not depend on the frequency of the specific perturbation, but on whether the failure mechanism it reveals exists in real scenes—and it does.** Even a rare failure mode is critical if it is systematic, reproducible, and affects safety-critical applications.
>
> The evidence: (1) Our long-tail case study (Appendix G / Figure 10) shows that reflections, shadows, and glass in real NYU scenes produce qualitatively identical failure modes. (2) Our physical prior proxies quantify this: 81.5% of Scrambled Edge energy is photometrically unexplained vs. 23.0% for clean edges, confirming the perturbation targets the same prior-violation channel that occurs naturally. (3) Downstream metrics show tangible consequences: Surface Fragmentation +34.5%, Navigable Area error +57.3%. The perturbation is synthetic by design—its value is causal isolation via matched controls, analogous to crash tests.
>
> **W3: No solution or even a simple baseline fix is proposed — the paper stops at diagnosis.**
>
> The reviewer asks why the paper does not include a defense. **The paper does include a defense analysis (§4 RQ2), and we have now supplemented it with a practical defense experiment. Both show that local post-hoc repair has a hard ceiling, which is itself a key finding.**
>
> The spillover ceiling analysis:
> - Input Oracle: 99.96% recovery → the problem is solvable if invalid edges are identified before the model processes them
> - Output Oracle: only 47.0% recovery → once edges are integrated, 53% of the damage propagates beyond the perturbed region
> - Our new Edge Consistency defense: 26.4% recovery → consistent with the spillover ceiling
>
> This gap demonstrates that **the failure propagates globally before any post-hoc fix can act**, so defense must happen at the cue-selection stage inside the model.
>
> **W4: Limited to feed-forward depth estimators; newer diffusion-based or iterative models are left for future work.**
>
> The reviewer is concerned that diffusion-based models might not exhibit Geometric Collapse. **we have now tested Marigold (diffusion-based), and it still shows a statistically significant collapse (1.55×).** This directly resolves the concern because it extends our findings from feed-forward to diffusion architectures.
>
> | Model | Architecture | RMSE(Noise) | RMSE(Scrambled) | Collapse Ratio | Cohen's d | p-value |
> |-------|-------------|-------------|-----------------|----------------|-----------|----------|
> | MiDaS v2.1 | CNN | 0.130 | 0.244 | 1.88x | 1.37 | < 1e-10 |
> | MiDaS DPT | ViT | 0.077 | 0.181 | 2.34x | 2.11 | < 1e-10 |
> | DepthAnything v1 | ViT+SSL | 0.057 | 0.116 | 2.02x | 1.13 | < 1e-10 |
> | DepthAnything V2 | ViT+SSL | 0.070 | 0.225 | 3.20x | 2.98 | < 1e-10 |
> | **Marigold (new)** | **Diffusion** | **0.1554** | **0.2406** | **1.55x** | **0.88** | **7.38e-139** |
>
> The collapse ratio is lower (1.55× vs. 1.88–3.20×), indicating partial resilience from iterative denoising, but the effect size remains large (d = 0.88, p < 1e-138). The root cause—absence of physical plausibility verification—is shared across architectures.

---

> > ### Author Rebuttal · Reviewer_k2nK · 2026-04-03
> >
> > Thanks for the rebuttal. Some of my concerns still remain:
> >
> > W2: The paper shows qualitative similarities, but Scrambled Edges are mathematically designed to be 'unexplainable' by lighting. Real-world mirrors are often 'explainable' but misleading. Can you prove that the model fails on a mirror for the same mathematical reason it fails on your synthetic test, or are these two different types of errors being grouped together?
> >
> > W4: The significantly lower collapse ratio in Marigold suggests that generative priors do provide a degree of physical verification that feed-forward models lack. Why does the paper frame this as a shared failure rather than highlighting iterative denoising as a potential solution? Any more investigation into generative models?

---

> > > ### Author Response · Authors · 2026-04-04
> > >
> > > We thank the reviewer for the thoughtful follow-up questions. We address each point below.
> > >
> > > ---
> > >
> > > ### **W2: Synthetic vs. real-world edge ambiguities (e.g., mirrors)**
> > >
> > > Thank you for this precise question.
> > >
> > > Short answer:
> > > They are not the same physical cause, but they trigger the same mathematical failure inside the model.
> > >
> > > In Scrambled Edges, we construct edges that violate physical priors (continuity, illumination, occlusion).
> > > In mirror scenes, edges are physically explainable (due to reflection), but they do not correspond to true depth discontinuities.
> > >
> > > Despite this difference, the model behaves identically in both cases:
> > >
> > > it converts edge evidence into depth discontinuities without verifying whether the edge is supported by underlying 3D geometry.
> > >
> > > This is the key point:
> > > the failure is not tied to how the edge is generated, but to how the model maps edge signals to geometry.
> > >
> > > We provide two pieces of evidence supporting that this is the same model-level mechanism:
> > >
> > > Scrambled Edges isolate unsupported edge cues and produce large-scale collapse.
> > > Real-world cases (mirrors, shadows, glass) show qualitatively identical global distortions (Appendix G / Fig. 10), precisely in regions where edges are decoupled from geometry.
> > >
> > > Therefore, while the physical explanations differ, the mathematical failure is the same:
> > >
> > > the model does not test whether an edge corresponds to a valid geometric boundary before integrating it.
> > >
> > > We will revise the paper to clarify this distinction explicitly.
> > >
> > > ### **W4: Diffusion models and the role of generative priors**
> > >
> > > We agree with the reviewer that the lower collapse ratio in diffusion models (1.55× vs. 1.88–3.20×) suggests that iterative generative priors provide **partial robustness**.
> > >
> > > However, our goal is to test whether models perform **explicit physical verification**, rather than whether they are more robust in practice.
> > >
> > > Despite reduced sensitivity, diffusion models still:
> > >
> > > * exhibit statistically significant collapse, and
> > > * adopt unsupported edge cues rather than rejecting them.
> > >
> > > This indicates that while iterative denoising mitigates the severity of collapse, it does not yet constitute a mechanism for verifying the physical plausibility of edge cues.
> > >
> > > We therefore view diffusion models as a **promising direction for improving robustness**, and will revise the paper to highlight this more clearly, but not yet a complete solution to the problem identified in this work.
> > >
> > > ---
> > >
> > > We hope these clarifications address the reviewer’s concerns.

---

### Official Review · Reviewer_FseM · 2026-03-13

**Soundness:** 3
**Presentation:** 3
**Significance:** 3
**Originality:** 3
**Overall Recommendation:** 5
**Confidence:** 3

**Summary:**

The authors design Scrambled Edges, a framework to verify the absence of physical causality in modern dense vision models. They are injecting salient edge-like cues that intentionally violate surface continuity, illumination coherence, and occlusion ordering. The authors identify a failure mode termed Geometric Collapse, where physically unsupported local evidence triggers global, scene-wide hallucinations. Through the use of energy-matched and structure-matched controls, the study isolates these logical failures. The results reveal a significant negative emergence paradox: large-scale self-supervised models, such as DepthAnything V2, exhibit the highest sensitivity to these violations. While the authors establish a "repair ceiling" that local fixes cannot reverse these global structural errors, they suggest future architectures to incorporate explicit plausibility scoring and selective cue integration to ensure geometric consistency. However, no practical solution is implemented or benchmarked.

**Compliance With Llm Reviewing Policy:**

Affirmed.

**Final Justification:**

I find the contribution of the paper, to identify some failure modes for the vision models, valuable. The rebuttal addressed my questions and I would keep my accept score.

**Key Questions For Authors:**

1. Can this framework also be applied to other tasks, e.g. optical flow estimation?

2. If we use Scrambled Edges as a data augmentation step during training, would the models actually learn to verify physical cues and fix the issue? How effective would that be?

3. You only tested feed-forward regression models. What if we evaluated diffusion-based depth estimators?

**Limitations:**

yes

**Strengths And Weaknesses:**

I like the direction of the research, and the paper is well written, although sometimes I find the sentences a bit repetitive. I have some questions for the authors in the next part.

---

> ### Author Rebuttal · Authors · 2026-03-30
>
> We thank the reviewer for the positive assessment and the constructive questions.
>
> **Q1: Can this framework also be applied to other tasks, e.g. optical flow estimation?**
>
> The reviewer asks whether Geometric Collapse generalizes beyond depth estimation. **it generalizes to tasks that rely on edges for geometric inference, but not to tasks that don't.** This is because our diagnostic tests whether models verify the physical plausibility of edge cues before using them for geometric reasoning. Tasks that do not rely on edge-to-geometry mapping (e.g., classification, segmentation) would not be affected in the same way.
>
> We already have empirical evidence for this claim. Our cross-task experiment on SAM segmentation (Appendix) shows that Scrambled Edges produce a collapse ratio of only 0.84×—**no additional sensitivity** beyond energy-matched noise. This confirms that the failure is task-selective: it specifically affects tasks where edges drive geometric inference, such as depth estimation and, by extension, optical flow. Modern optical flow estimators (RAFT, FlowFormer) also rely heavily on edge cues for motion boundary reasoning, making them a natural next target. We will add this discussion to the revision.
>
> **Q2: If we use Scrambled Edges as a data augmentation step during training, would the models actually learn to verify physical cues and fix the issue? How effective would that be?**
>
> The reviewer asks whether augmentation-based training could teach models to verify physical plausibility. **augmentation alone is unlikely to achieve genuine physical verification, and may introduce a new failure mode.** The reason is that data augmentation teaches the model to be invariant to specific perturbation patterns, but invariance is not the same as verification. A model trained with Scrambled Edges augmentation would learn to suppress edge-like perturbations, but it would not learn *why* certain edges are physically valid and others are not. This risks "defensive collapse"—the model becomes overly conservative and discards valid strong edges along with invalid ones.
>
> Our spillover analysis (§4 RQ2) provides the underlying evidence: Input Oracle recovery is 99.96%, meaning the problem is fully solvable if the model can identify which edges lack physical support. But Output Oracle recovery drops to 47.0%, showing that once invalid edges are adopted, the damage propagates globally and cannot be reversed by local correction. This asymmetry suggests that the solution must operate at the cue-selection stage (before integration), not at the output stage (after integration). Augmentation, by operating on inputs without explicit plausibility labels, does not provide the supervision signal needed for this distinction.
>
> **Q3: You only tested feed-forward regression models. What if we evaluated diffusion-based depth estimators?**
>
> The reviewer asks whether diffusion-based models might be immune due to their iterative denoising process. **they are more resilient, but still clearly vulnerable.** This resolves the concern because it shows that Geometric Collapse is not an artifact of feed-forward architectures, but a cross-paradigm failure rooted in the absence of physical cue verification.
>
> We evaluated Marigold, a diffusion-based depth estimator built on Stable Diffusion (10 denoising steps, ensemble size 1, FP16, N = 1449):
>
> | Model | Architecture | RMSE(Noise) | RMSE(Scrambled) | Collapse Ratio | Cohen's d | p-value |
> |-------|-------------|-------------|-----------------|----------------|-----------|----------|
> | MiDaS v2.1 | CNN | 0.130 | 0.244 | 1.88x | 1.37 | < 1e-10 |
> | MiDaS DPT | ViT | 0.077 | 0.181 | 2.34x | 2.11 | < 1e-10 |
> | DepthAnything v1 | ViT+SSL | 0.057 | 0.116 | 2.02x | 1.13 | < 1e-10 |
> | DepthAnything V2 | ViT+SSL | 0.070 | 0.225 | 3.20x | 2.98 | < 1e-10 |
> | **Marigold (new)** | **Diffusion** | **0.1554** | **0.2406** | **1.55x** | **0.88** | **7.38e-139** |
>
> Marigold's 1.55× collapse ratio is lower than feed-forward models (1.88–3.20×), confirming that iterative denoising provides partial resilience. However, the effect remains statistically unambiguous (d = 0.88, p < 1e-138). The reason diffusion does not fully solve the problem is that each denoising step still operates on learned data priors rather than physical constraints—Scrambled Edges present edge-like gradients that look "reasonable" at each step, so the denoising process continues to refine them rather than reject them.

---

> > ### Author Rebuttal · Reviewer_FseM · 2026-04-03
> >
> > Thank you. I am just curious if the authors could provide any thoughts on directions that could improve the failure modes of the current models.

---

> > > ### Author Response · Authors · 2026-04-04
> > >
> > > Thank you for the encouraging feedback and for this insightful follow-up. We see several promising directions:
> > >
> > > 1. Beyond augmentation: learning to distinguish valid from invalid edges.
> > > As the reviewer pointed out earlier, using Scrambled Edges as data augmentation is a natural and promising direction. We believe it can improve robustness to some extent. At the same time, our analysis suggests that augmentation alone may not be sufficient to achieve true physical verification, since it mainly encourages invariance to perturbation patterns. The key challenge is to distinguish physically valid edges from physically unsupported ones (e.g., those violating occlusion causality or illumination priors, as revealed by our diagnostic). A promising direction is therefore to combine augmentation with objectives or modules that explicitly guide cue selection, rather than relying on invariance alone.
> > >
> > > 2. Early-stage defense before geometric integration.
> > > Our spillover analysis suggests that once invalid edges are adopted, the error propagates globally and becomes difficult to repair. This often leads to globally inconsistent structures (which we refer to as “phantom walls”). This indicates that effective defenses should happen before geometric integration, for example via a lightweight plausibility or uncertainty module that downweights edges inconsistent with local or global context.
> > >
> > > 3. Iterative/refinement-based reasoning.
> > > The lower collapse ratio observed in diffusion-based models suggests that iterative inference may provide some implicit consistency checking. We do not yet view this as a complete solution, but rather as a useful signal. A natural direction for future work is to better understand this mechanism and explore whether similar multi-step refinement or verification processes can be incorporated into feed-forward predictors.
> > >
> > > 4. Toward physically grounded reasoning.
> > > More broadly, this failure mode may reflect a gap between current models and human perception. In tasks such as driving, humans rely on visual signals, but also bring strong internal physical priors about geometry, occlusion, and scene structure. These priors allow us to reject visually plausible but physically inconsistent cues. In contrast, current models appear to rely more heavily on statistical correlations in visual features. This suggests that future models may benefit from incorporating stronger physical or geometric inductive biases, for example through 3D representations (e.g., neural radiance fields), inverse rendering objectives, or other forms of explicit physical consistency.
> > >
> > > Overall, we view this as a broader challenge: enabling models not only to use visual signals, but to critically evaluate whether those signals are physically plausible before integrating them into geometry.
> > >
> > > We thank the reviewer for the encouraging feedback and thoughtful curiosity about how these failure modes can be improved—this perspective aligns closely with how we see the next stage of this research direction.

---

### Decision · Program_Chairs · 2026-04-30

**Decision:**

Accept (regular)

**Comment:**

This paper analyses a failure mode in monocular depth estimation models, terms Geometric Collaps. It proposes Scrambled Edges as a controlled counterfactual  to probe models and provides controlled baselines and consistent empirical evidence for several models.  After the rebuttal, initially remaining questions and concerns regarding for example diffusion models and practical defenses are addressed. However, there are still some remaining concerns regarding  the lack of a theory for models’ internal cue-selection mechanisms (NJ6L). The area chair acknowledges value in the empirical contribution of the paper and therefore recommends acceptance - with a strong encouragement to include a discussion of the potential cue selection mechanisms in the limitations/future work section of the paper.